**Data Availability Statement:** The reported data underlying the results presented in the study are from an open database deposited in the literature

# Modeling the influence of lime on the unconfined compressive strength of reconstituted graded soil using advanced machine learning approaches for subgrade and liner applications

Xinghuang Guo[1], Cesar Garcia[2], Alexis Ivan Andrade Valle[3,4], Kennedy Onyelowe[5,6,7]*, Andrea Natali Zarate Villacres[8], Ahmed M. Ebid[9], Shadi Hanandeh[10]

1 China Design Group, Nanjing City, Jiangsu Province, China, 2 Facultad de Ingenieria, Arquitectura, Universidad Nacional de Chimborazo (UNACH), Riobamba, Ecuador, 3 Facultad de Ingenieria, Ingenieria Civil, Universidad Nacional de Chimborazo (UNACH), Riobamba, Ecuador, 4 PhD Scholar, Architecture, Heritage and the City, Universitat Politecnica de Valencia, Valencia, Espana, 5 Department of Civil Engineering, Michael Okpara University of Agriculture, Umudike, Nigeria, 6 Department of Civil Engineering, University of the Peloponnese, Patras, Greece, 7 Department of Civil Engineering, Kampala International University, Kampala, Uganda, 8 Facultad de Ingenieria, Ingenieria Civil, Universidad Nacional de Chimborazo, Riobamba, Chimborazo, Ecuador, 9 Department of Civil Engineering, Future University in Egypt, New Cairo, Egypt, 10 Civil Engineering Department, Al-Balqa Applied University, As-Salt, Jordan

* konyelowe@mouau.edu.ng, k.onyelowe@go.uop.gr, kennedychibuzor@kiu.ac.ug

## Abstract

In the field of soil mechanics, especially in transportation and environmental geotechnics, the use of machine learning (ML) techniques has emerged as a powerful tool for predicting and understanding the compressive strength behavior of soils especially graded ones. This is to overcome the sophisticated equipment, laboratory space and cost needs utilized in multiple experiments on the treatment of soils for environmental geotechnics systems. This present study explores the application of machine learning (ML) techniques, namely Genetic Programming (GP), Artificial Neural Networks (ANN), Evolutionary Polynomial Regression (EPR), and the Response Surface Methodology in predicting the unconfined compressive strength (UCS) of soil-lime mixtures. This was for purposes of subgrade and landfill liner design and construction. By utilizing input variables such as Gravel, Sand, Silt, Clay, and Lime contents (G, S, M, C, L), the models forecasted the strength values after 7 and 28 days of curing. The accuracy of the developed models was compared, revealing that both ANN and EPR achieved a similar level of accuracy for UCS after 7 days, while the GP model performed slightly lower. The complexity of the formula required for predicting UCS after 28 days resulted in decreased accuracy. The ANN and EPR models achieved accuracies of 85% and 82%, with $R^2$ of 0.947 and 0.923, and average error of 0.15 and 0.18, respectively, while the GP model exhibited a lower accuracy of 66.0%. Conversely, the RSM produced models for the UCS with predicted $R^2$ of more than 98% and 99%, for the 7- and 28- day curing regimes, respectively. The RSM also produced adequate precision in modelling UCS of more than 14% against the standard 7%. All input factors were found to have almost equal

"64. Gajurel, Amit; Chittoori, Bhaskar; Mukherjee, Partha Sarathi; Sadegh, Mojtaba, 2021, "Database used in Machine Learning Methods to Map Stabilizer Effectiveness based on Common Soil Properties", https://doi.org/10.18738/T8/QORZZZ, Texas Data Repository, V3" which has been cited in the intext as [64] and listed accordingly and it has been openly reported in the manuscript as Table 1. There are no permissions required prior to its use as it is openly available in the literature and is being used openly.

**Funding:** The author(s) received no specific funding for this work.

**Competing interests:** The authors have declared that no competing interests exist.

importance, except for the lime content (L), which had an average influence. This shows the importance of soil gradation in the design and construction of subgrade and landfill liners. This research further demonstrates the potential of ML techniques for predicting the strength of lime reconstituted G-S-M-C graded soils and provides valuable insights for engineering applications in exact and sustainable subgrade and liner designs, construction and performance monitoring and rehabilitation of the constructed civil engineering infrastructure.

## 1. Introduction

Soil stabilization is the process of changing or maintaining one or more soil qualities to improve a soil's engineering features and performance [1]. This process is executed during the construction of civil engineering infrastructures like pavement foundation (e.g., subgrade) and landfill liners (compacted earth liners). The technique of adding a specific soil, chemical components, or other cementing material to a natural soil to improve one or more of its attributes is known as soil stabilization. Stabilization can be achieved by mechanically mixing natural soil and stabilizing material together to form a homogeneous mixture, or by applying the stabilizing substance to an undisturbed soil deposit and allowing it to infiltrate through soil voids to achieve interaction [2]. Stabilization is a 5000-year-old notion [3] stated that ancient Egypt and Mesopotamia used stabilized earth roads and that the Greeks and Romans used lime as a stabilizer. The addition of cement or lime to soil can help to stabilize it. Such stabilization procedures improve the stabilized soil's varied engineering qualities, resulting in better construction material. The benefits of soil stabilization include increased soil strength, durability stiffness, changes in permeability, density, porosity, and volume, waterproofing, reduced surface abrasion, and reduced soil plasticity and swelling/shrinkage potential [4,5]. Lime is frequently used as a soil stabilizing agent because it is readily accessible, inexpensive, and effective at enhancing the soil's strength and durability. The unconfined compressive strength, abbreviated as UCS, is frequently used to determine the efficacy of different soil stabilization techniques [6]. Apart from other qualities of stabilized soil, scientists generally agree that the most important outcome of stabilization is unconfined compressive strength (UCS) [7]. The stabilized soil's unconfined compressive strength is determined by the water and the cement amount in the mixture, the properties and types of the soil, and the curing and mixing conditions [8–12]. The UCS determination of stabilized soil is critical in the construction and improvement design. Some research has been done to forecast the UCS of stabilized soil using input parameters including the binder concentration, water/binder ratio, and curing duration [13–17].

Many researchers have used soft computation and other techniques to develop models for estimating UCS values of stabilized soils [1,15,18–20]. In recent decades, computational intelligence methods that are accomplished of estimating the input-output non-linear relationships for many complicated issues have piqued interest [21]. As previously stated, the UCS is influenced by several factors. Determining the UCS of soil samples requires time-consuming and labor-intensive lab work. Developing predictive models to deal with this problem could be advantageous. Many methods, such as traditional linear regressions, can be used to develop such behavioral models [22,23]. However, regression techniques have several important problems, such as expecting a pre-specified nonlinear or linear relationship between the outputs and inputs, which is not necessarily the case [24]. Machine learning (ML) techniques have been more well-liked in recent years as a result of their capacity to assess and anticipate

intricate relationships between output and input variables. This ability has helped the techniques become more widely used. Three of the most popular machine learning methods used for this are Gaussian processes (GP), evolutionary programming (EPR), and artificial neural networks (ANN). GP is a method based on probability that employs a kernel function to model the relationship between input variables and output variables. It has been effectively implemented in a variety of applications, containing soil mechanics, to model the behavior of soil and predict its engineering properties. EPR is a method for population-based optimization that imitates the process of natural evolution in order to discover the optimum solution to a problem. In the discipline of soil mechanics, it has been used to enhance the design of foundations and to simulate the behavior of soil under a variety of loading conditions [25].

An Artificial Neural Network (ANN) is a computational model made up of various procedures modules in which neurons generate output values depending on the input values. ANN is getting popular in a variety of geotechnical engineering applications [26,27]. Some researches, such as reported, [14,15,28–31] have used ANN models to estimate the UCS value in chemical soil stabilization. The behavior of soil and its engineering properties can be predicted using ANN, which has found applications in a variety of fields, including soil mechanics. Analytical formulas depend on ANN models are more precise than multivariable nonlinear regression or high-performance multiple regression analysis. ANN, on the other hand, is regarded as a "black box" application. Complex transfer functions, such as logistic sigmoid and hyperbolic tangent sigmoid functions, are used to create prediction equations. As an outcome, the use of ANN-based prediction purposes is limited because they cannot easily calculate the output using the input values [32]. As a result, in this study, the ANN approach is used to estimate a UCS prediction model. Despite the ANN models' reasonable performance, they need practical equations for calculating the results. Pharm et al. [27] used the ANN approach to develop the UCS prediction model concerning the specified parameters. A statistical analysis reveals that the suggested model created in this work is dependable and accurate, with a high relationship coefficient and minimal root mean squared errors. The most important variables impacting the UCS value, according to the ANN-based model, are and cement content the soil particles percentage passing filter 0.5 mm. Jahed Armaghoni et al. [33] developed three models to forecast UCS of granite: multiple regression analysis (MRA), adaptive neuro-fuzzy inference system (ANFIS), and artificial neural network (ANN). ANFIS is more precise than the other two models, they discovered. According to the test data, it is also proven that ANN outperforms MRA. Sharma et al [34] established four numerous and simple linear regression models to forecast UCS of structured artificial soil. All of their models have an $R^2$ of greater than 0.9, indicating that they are accurate. More sophisticated models with more input variables, on the other hand, showed greater accuracy.

Genetic programming (GP) is a type of managed machine learning approach based on Darwin's theory of evolution [35]. It's a different way of looking at behavior modeling. Gene-expression programming (GEP) is a division of GP that uses a computer program to generate a solution to a problem [36], and it is the method most typically employed in geotechnical engineering [37]. GEP also selects populations, depending on fitness purpose and presents them with a gene via numerous operators [38]. Without making any expectations about the likely functional connections structure, GEP can construct strong prediction functions [39]. The GEP model is a reliable, strong, and precise forecasting method. Furthermore, GEP-based equations are more practical and transparent than ANN-based formulas. As a result, the predictive proposed equations derived from the GEP model may be ready for use.

Cement, asphalt, and lime were combined with weak soil in this study to rise the strength of the soil specimens. The impacts of additive contents (i.e. cement, asphalt, and lime) on the UCS value were investigated using UCS tests on stabilized specimens. The effective majors are

also used to estimate the UCS value using MLR and NLR methods. The prediction formulas among variables and the relative relevance of input parameters were measured using ANNs and GAs analysis. The UCS data obtained from soaked and unsoaked mixtures were combined in this study. This means that the UCS values obtained from the soil-lime mixture after it had been soaked were combined with the UCS values obtained from the mixture before it had been soaked. This method is frequently utilized in soil mechanics research in order to provide a more in-depth investigation of the efficiency of various soil stabilization strategies. In order to assess the stocked UCS's contribution to the improvement of the lime reconstituted mixture, the ML techniques of GP, EPR, and ANN were utilized. The behavior of soil is highly complex and nonlinear, and traditional analytical and empirical approaches may not be able to capture all of the underlying relationships between output and input variables. Because of this, the usage of ML techniques is particularly useful in soil mechanics [40].

GP, EPR, and ANN can all model complex relationships and make predictions based on the input data. GP is a non-parametric method that suppose no functional particular form for the correlation between output and input variables. Therefore, it is capable of capturing nonlinear relationships and managing chaotic data. EPR is a population-based optimization method that can seek a vast search space for the optimal solution. This capability makes EPR a useful modeling tool for complex systems [41]. Due to its adaptability, ANN has been widely employed to model intricate relationships in numerous fields, including soil mechanics. The stored UCS data for the soil-lime mélange served as input data for machine learning techniques, and ML models were trained using the input data to predict the UCS values. The ML methodologies utilized the accumulated UCS data for the soil-lime mixture as input data. After instructing the models, they were used to determine if the addition of calcium to the soil increased UCS values [42].

The use of both unsoaked and soaked UCS data in the study allows for a more comprehensive evaluation of the soil-lime mixture's efficacy, as the soaked UCS values are more indicative of the mixture's performance in moist conditions. This is due to the fact that the unsoaked UCS values reflect the mixture's performance in dry conditions [43].

ML methods can also be employed to optimize soil stabilization systems by identifying the optimal combination of stabilizing compounds, soil varieties, and other variables. This can be accomplished by identifying the optimal combination of factors. Engineers can use this information to design systems that are more effective, efficient, and cost-effective, while still meeting specific performance requirements [44]. However, there are some limitations to the application of ML methods in soil mechanics. To train machine learning models, high-quality data is required, which can be considered a drawback of the technology. In the field of soil mechanics, where data acquisition can be time-consuming and costly, this can be challenging. Moreover, ML models can be challenging to interpret, making it difficult to comprehend the underlying mechanisms that regulate soil behavior [45]. This can make understanding the mechanisms underlying soil behavior more difficult. Overall, ML techniques provide a robust and adaptable method for evaluating the effectiveness of soil stabilization techniques, and they can be combined with other techniques to provide a more complete understanding of soil behavior. Additionally, these methodologies can be used to evaluate the effectiveness of soil stabilization techniques [46].

The application of ML techniques in this study provides an essential tool for evaluating the effectiveness of soil stabilization techniques and can aid engineers in designing more efficient and effective soil stabilization systems. In addition to the previously mentioned ML techniques, additional methods exist for evaluating the effectiveness of soil stabilization techniques [47]. Techniques that fall into this category contain empirical, analytical, and numerical approaches [48]. Empirical methods are frequently used in the field of soil mechanics to

develop correlations between the engineering properties of soil and other factors such as soil type, moisture content, and confining pressure [49]. Statistical analysis of experimental data forms the basis for these empirical methodologies [12]. Despite providing estimates of soil behavior in a fast and straightforward manner, it is possible that these techniques are limited in their ability to capture complex relationships [41,50].

Analytical methods are founded on mathematical models that characterize the behavior of soil using equations derived from fundamental principles of soil mechanics [51]. These mathematical models provide the foundation for analytical methods [52]. These methodologies can enhance the design of soil stabilization systems by providing a deeper understanding of soil behavior. However, in order to simplify them, it may be necessary to make assumptions that reduce their precision [53]. The foundation of numerical methods is the use of numerical models, such as finite element analysis (FEA) and discrete element method (DEM), in computer simulations of soil behavior [54]. These methods are supported by computer simulations [48]. Nevertheless, these methods can be computationally intensive and require a substantial amount of knowledge to be utilized effectively [55]. In contrast, they can provide precise and exhaustive predictions of soil behavior. When making a decision regarding soil stabilization techniques, it is essential to consider the specific application, the available data, the desired level of precision and complexity, and the evaluation method [56]. The implementation of ML techniques offers a method that is both robust and adaptable, and it can be combined with other techniques to provide a more comprehensive analysis of the behavior of soil.

In the discipline of soil mechanics, techniques based on machine learning have several advantages over more conventional methods. In the field of soil mechanics, where data can be chaotic and highly variable, their ability to manage large and complex datasets is especially advantageous [57–59]. Traditional methods may not be able to identify nonlinear relationships between input and output variables, whereas ML techniques can [58]. The ability of machine learning techniques to learn from data and adapt to changing conditions is an additional advantage of using these methods [57]. This is particularly useful in the field of soil mechanics, which is significant because the behavior of soil is highly dependent on a number of variables, including its moisture content, confining pressure, and loading rates. Machine learning techniques can learn from data collected under diverse conditions and predict future conditions [47].

In this research paper, the strengths at 7- and 28-days curing period have been forecasted by using four intelligent techniques for a reconstituted soil treated with lime at different dosages. This is for the purpose of subgrade (as a transportation geotechnic structure) and landfill liner (as an environmental geotechnic infrastructure) design and construction. These structures have special benefits they offer to the overall life of humanity according to the UNSDGs. Transportation geotechnics and environmental geotechnics are two specialized branches of geotechnical engineering that focus on the application of soil mechanics, rock mechanics, and other geotechnical principles to the design, construction, and maintenance of transportation infrastructure and environmental projects [15]. Transportation geotechnics involves the application of geotechnical engineering principles to the design and construction of transportation infrastructure such as highways, railways, airports, and ports. Subgrade and pavement design: This involves evaluating the properties of the underlying soil or rock to ensure that it can support the loads imposed by traffic and environmental conditions, and designing suitable pavement structures. Slope stability and embankment design: Assessing the stability of natural and man-made slopes, as well as designing and constructing embankments that can support transportation infrastructure [17]. Foundation design for transportation structures: Ensuring that transportation structures such as bridges, tunnels, and retaining walls have stable foundations in a variety of soil and rock conditions. Ground improvement techniques: Implementing

methods to strengthen or stabilize the ground, such as soil compaction, reinforcement, grouting, or soil replacement, to improve the performance of transportation infrastructure [35]. On the other hand, environmental geotechnics focuses on the application of geotechnical engineering principles to address environmental challenges, such as waste management, contaminated site remediation, and sustainable infrastructure development [23]. Landfill design and management: Evaluating the geotechnical properties of the site for landfill construction, designing liners and leachate collection systems to prevent environmental contamination, and managing the stability and settlement of waste fills. Contaminated site remediation: Applying geotechnical techniques to remediate sites contaminated with pollutants, such as heavy metals, hydrocarbons, or hazardous chemicals, by using methods like soil stabilization, containment, and in-situ treatment. Geotechnical aspects of sustainable infrastructure: Incorporating geotechnical considerations into the design and construction of sustainable infrastructure, such as green roofs, permeable pavements, and engineered natural systems for stormwater management [42]. Both transportation geotechnics and environmental geotechnics require a thorough understanding of soil behavior, groundwater conditions, and the interaction between soil and man-made structures [26]. Many more references have been made that studied the use lime in single and or combination of other cementitious materials such as the application of cement/ lime combination with pozzolans accompanied with crushed stone waste to improve the strength of soil [60], the application of lime-nano-silica combination [61], the treatment of silty soil with lime considering the strength improvement comparison between UCS and splitting tensile strength [62], and the application of bagasse-lime combination, where ANN was also utilized to model the behavior of this material combination in soil [63]. Yet, none tried the use of the novel response surface methodology (RSM) in the combination of other regression techniques to model these problems. Also, the application of lime has continued to come in the combination of other materials. But, in the present work, lime has been applied as the only cementitious material and a combination of four machine learning techniques have been used to predict the strength development of the treated soil considering its gradation pattern. Geotechnical engineers working in these fields play a crucial role in ensuring the safe, cost-effective, and sustainable development of transportation and environmental infrastructure. For the need to solve most environmental geotechnics problems, the more pronounced of which are those related to landfills and pavement foundations across the world, this work has been undertaken to propose more reliable and robust mathematical models based on advanced machine learning techniques. These techniques are flexible because of the advantage of utilizing proposed closed-form equations to apply the models manually as well as smartly. Landfills and pavement subgrade systems require immediate attention from design to construction and to usage over the period to monitor the optimized utilization of the sustainable materials used in their construction. So, this research project presents a potential for use in this area. Many other research works are stated above have been carried but none presents a combination of the four techniques applied in this paper. Meanwhile, the flowchart of the present research project is illustrated in Fig 1.

## 2. Data collection and statistical analysis

The complete database constitutes of an open database of 136 records, which were collected from experimentally tested samples of soil stabilized with lime deposited at the US soil stabilization database, which can be found in the cited literature [64]. These soils were collected across the world especially the United States in the East central Iowa, North central Florida, Illinois, Kansas, etc. and classified as dune sand, kaolinite clay, illinite clay, montmorillonite clay, alluvial, sand loess, friable loess, plastic loess, leached Kansas till, unleached Kansas till,

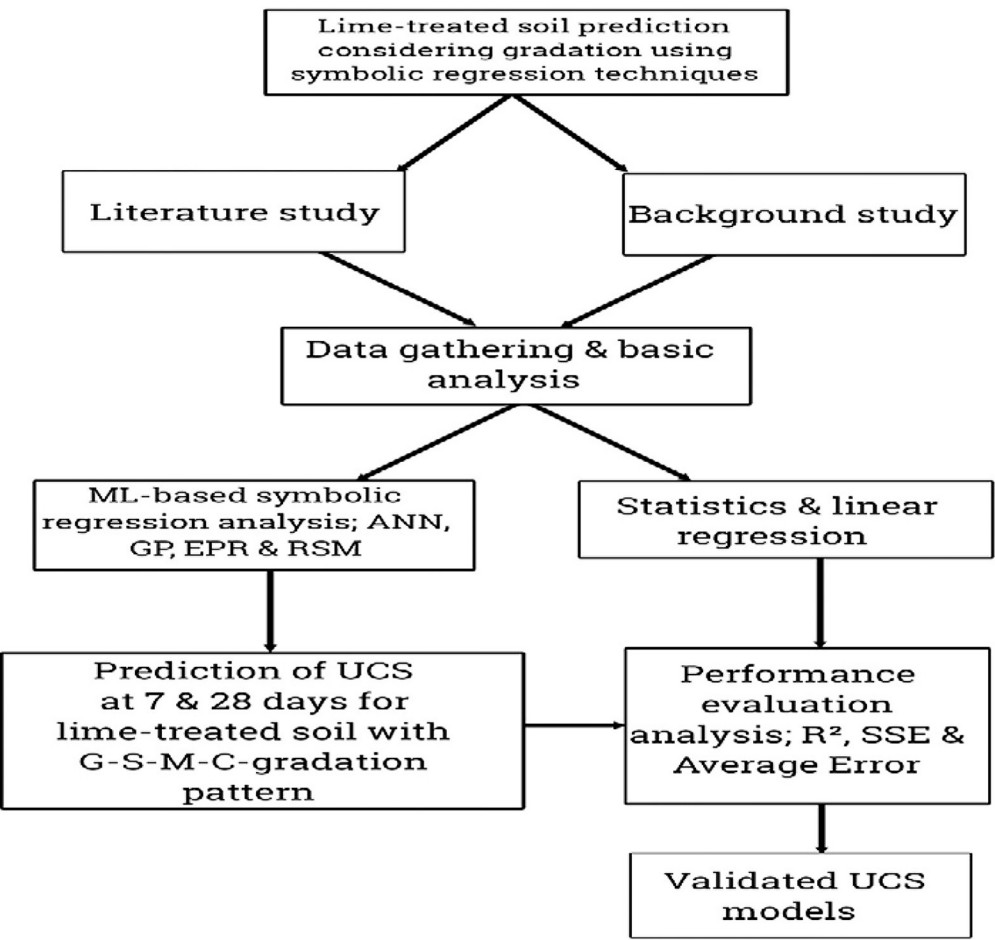

**Fig 1. The flowchart of the research project.**

etc. [64]. The lime reconstituted graded soil samples were collected at different locations within US soil and other points within the upper greater Europe giving a global representation of soils. Each record contains the following data:

- Gravel content, G = 100.G/ (G+S+M+C)

- Sand content, S = 100.S/ (G+S+M+C)

- Silt content, M = 100.M/ (G+S+M+C)

- Clay content, C = 100.C/ (G+S+M+C)

- Lime dose, L = 100.L/ (G+S+M+C)

- Unconfined compressive strength after 7 days (kPa), UCS7

- Unconfined compressive strength after 28 days (kPa), UCS28

The collected records were partitioned into a training set of $\approx 75\%$ (100 records) and validation set of $\approx 25\%$ (36 records) observing the conditions of the k-fold cross-validation to overcome under-fitting issues in the model. Tables 1 and 2 include the complete dataset for the training and validation respectively, while Tables 3 and 4 summarize their statistical

**Table 1. Training dataset.**

| G | S | M | C | L | UCS7 | UCS28 | G | S | M | C | L | UCS7 | UCS28 |
|---|---|---|---|---|---|---|---|---|---|---|---|---|---|
| % | % | % | % | % | kPa | kPa | % | % | % | % | % | kPa | kPa |
| | | | | | | **Training set** | | | | | | | |
| 0 | 11 | 39 | 48 | 2 | 272 | 217 | 0 | 10 | 38 | 52 | 8 | 1116 | 1707 |
| 10 | 41 | 30 | 18 | 4 | 1760 | 3461 | 1 | 29 | 33 | 35 | 8 | 892 | 1387 |
| 1 | 0 | 57 | 42 | 8 | 1028 | 1843 | 1 | 29 | 33 | 35 | 4 | 952 | 914 |
| 0 | 0 | 60 | 39 | 12 | 948 | 1667 | 0 | 31 | 30 | 38 | 6 | 1200 | 1818 |
| 0 | 3 | 36 | 61 | 4 | 778 | 925 | 10 | 11 | 37 | 42 | 4 | 372 | 411 |
| 0 | 6 | 18 | 75 | 4 | 729 | 1079 | 0 | 31 | 30 | 38 | 12 | 1264 | 2283 |
| 10 | 11 | 37 | 42 | 8 | 545 | 840 | 0 | 1 | 56 | 41 | 2 | 157 | 164 |
| 1 | 0 | 70 | 29 | 4 | 890 | 1756 | 0 | 0 | 60 | 39 | 10 | 866 | 1566 |
| 0 | 31 | 30 | 38 | 10 | 1318 | 2013 | 1 | 0 | 60 | 39 | 4 | 754 | 1332 |
| 1 | 21 | 15 | 62 | 12 | 1166 | 1593 | 8 | 26 | 33 | 32 | 8 | 1313 | 2834 |
| 0 | 0 | 81 | 19 | 6 | 1392 | 2758 | 0 | 2 | 28 | 68 | 2 | 174 | 195 |
| 10 | 41 | 30 | 18 | 12 | 1409 | 2842 | 0 | 0 | 60 | 39 | 14 | 913 | 1729 |
| 0 | 11 | 39 | 48 | 12 | 906 | 1533 | 0 | 38 | 47 | 12 | 4 | 233 | 302 |
| 0 | 2 | 28 | 68 | 8 | 1170 | 2095 | 10 | 11 | 37 | 42 | 12 | 600 | 1251 |
| 0 | 31 | 30 | 38 | 4 | 1127 | 1901 | 9 | 29 | 31 | 30 | 12 | 1696 | 2941 |
| 1 | 29 | 33 | 35 | 12 | 832 | 1305 | 1 | 0 | 60 | 39 | 12 | 1061 | 2536 |
| 0 | 3 | 36 | 61 | 10 | 1250 | 2217 | 0 | 0 | 60 | 39 | 4 | 309 | 873 |
| 0 | 6 | 18 | 75 | 6 | 923 | 1427 | 0 | 0 | 52 | 46 | 8 | 894 | 1643 |
| 0 | 38 | 47 | 12 | 8 | 277 | 375 | 0 | 0 | 60 | 39 | 8 | 755 | 1445 |
| 0 | 1 | 56 | 41 | 8 | 341 | 484 | 0 | 37 | 38 | 23 | 2 | 175 | 147 |
| 0 | 1 | 61 | 36 | 12 | 372 | 501 | 0 | 16 | 13 | 70 | 8 | 917 | 1379 |
| 1 | 0 | 60 | 39 | 10 | 1037 | 2313 | 10 | 11 | 37 | 42 | 10 | 581 | 1054 |
| 0 | 20 | 40 | 38 | 12 | 1910 | 3038 | 0 | 0 | 81 | 19 | 2 | 802 | 417 |
| 0 | 2 | 69 | 28 | 4 | 152 | 145 | 3 | 33 | 30 | 33 | 8 | 902 | 1059 |
| 0 | 1 | 69 | 29 | 8 | 420 | 683 | 1 | 45 | 18 | 36 | 12 | 1085 | 2248 |
| 0 | 0 | 81 | 19 | 12 | 1628 | 2924 | 0 | 38 | 47 | 12 | 12 | 405 | 566 |
| 0 | 20 | 40 | 38 | 4 | 1745 | 2839 | 0 | 7 | 36 | 57 | 4 | 952 | 1611 |
| 0 | 16 | 13 | 70 | 10 | 1108 | 1493 | 0 | 0 | 52 | 46 | 4 | 534 | 882 |
| 0 | 3 | 36 | 61 | 12 | 1344 | 2536 | 0 | 0 | 68 | 31 | 12 | 700 | 1224 |
| 0 | 7 | 36 | 57 | 8 | 893 | 1557 | 0 | 0 | 60 | 39 | 2 | 140 | 91 |
| 0 | 7 | 36 | 57 | 6 | 964 | 1564 | 0 | 18 | 42 | 39 | 2 | 127 | 120 |
| 0 | 2 | 69 | 28 | 8 | 284 | 277 | 10 | 11 | 37 | 42 | 6 | 458 | 609 |
| 0 | 10 | 38 | 52 | 6 | 1187 | 1864 | 0 | 31 | 30 | 38 | 2 | 397 | 230 |
| 1 | 45 | 18 | 36 | 6 | 1026 | 2231 | 0 | 38 | 47 | 12 | 2 | 182 | 237 |
| 3 | 33 | 30 | 33 | 2 | 488 | 712 | 1 | 0 | 60 | 39 | 8 | 990 | 2185 |
| 1 | 21 | 15 | 62 | 8 | 882 | 1286 | 0 | 1 | 69 | 29 | 2 | 235 | 163 |
| 0 | 11 | 39 | 48 | 8 | 792 | 1398 | 0 | 16 | 13 | 70 | 12 | 1231 | 1923 |
| 9 | 29 | 31 | 30 | 8 | 1843 | 4219 | 0 | 1 | 61 | 36 | 8 | 303 | 439 |
| 3 | 33 | 30 | 33 | 12 | 895 | 1163 | 0 | 6 | 18 | 75 | 12 | 876 | 1339 |
| 8 | 26 | 33 | 32 | 12 | 1298 | 2036 | 10 | 41 | 30 | 18 | 8 | 1663 | 3207 |
| 0 | 2 | 28 | 68 | 12 | 1201 | 2208 | 9 | 29 | 31 | 30 | 4 | 2141 | 3883 |
| 0 | 1 | 61 | 36 | 2 | 177 | 170 | 1 | 0 | 60 | 39 | 6 | 967 | 2034 |
| 1 | 21 | 15 | 62 | 4 | 329 | 329 | 0 | 20 | 40 | 38 | 6 | 2193 | 3757 |
| 1 | 0 | 57 | 42 | 4 | 861 | 1398 | 0 | 3 | 36 | 61 | 8 | 1155 | 1875 |
| 0 | 20 | 40 | 38 | 10 | 2335 | 3557 | 1 | 0 | 70 | 29 | 2 | 393 | 150 |
| 0 | 0 | 60 | 39 | 6 | 527 | 1538 | 0 | 16 | 13 | 70 | 12 | 1232 | 1693 |
| 1 | 45 | 18 | 36 | 10 | 1082 | 2406 | 0 | 37 | 38 | 23 | 12 | 278 | 313 |
| 1 | 29 | 33 | 35 | 2 | 656 | 858 | 0 | 31 | 30 | 38 | 8 | 1283 | 2175 |

*(Continued)*

**Table 1.** (Continued)

| G | S | M | C | L | UCS7 | UCS28 | G | S | M | C | L | UCS7 | UCS28 |
|---|---|---|---|---|---|---|---|---|---|---|---|---|---|
| % | % | % | % | % | kPa | kPa | % | % | % | % | % | kPa | kPa |
| 0 | 0 | 68 | 31 | 8 | 714 | 1047 | 0 | 37 | 38 | 23 | 8 | 248 | 304 |
| 0 | 0 | 81 | 19 | 4 | 1184 | 2691 | 0 | 0 | 68 | 31 | 4 | 573 | 941 |

**Table 2. Validation dataset.**

| G | S | M | C | L | UCS7 | UCS28 |
|---|---|---|---|---|---|---|
| % | % | % | % | % | kPa | kPa |
| Validation set | | | | | | |
| 0 | 7 | 36 | 57 | 10 | 919 | 1591 |
| 0 | 1 | 61 | 36 | 4 | 291 | 413 |
| 1 | 45 | 18 | 36 | 8 | 1029 | 2328 |
| 9 | 29 | 31 | 30 | 2 | 1217 | 1285 |
| 0 | 16 | 13 | 70 | 6 | 572 | 457 |
| 0 | 20 | 40 | 38 | 8 | 2217 | 3764 |
| 0 | 2 | 69 | 28 | 12 | 286 | 365 |
| 0 | 10 | 38 | 52 | 10 | 1100 | 1618 |
| 0 | 3 | 36 | 61 | 6 | 1014 | 1555 |
| 8 | 26 | 33 | 32 | 2 | 1380 | 2320 |
| 0 | 2 | 28 | 68 | 4 | 739 | 852 |
| 0 | 18 | 42 | 39 | 8 | 215 | 377 |
| 0 | 7 | 36 | 57 | 12 | 930 | 1598 |
| 10 | 41 | 30 | 18 | 2 | 1406 | 1696 |
| 0 | 18 | 42 | 39 | 12 | 302 | 443 |
| 1 | 45 | 18 | 36 | 4 | 1007 | 1696 |
| 0 | 1 | 56 | 41 | 4 | 252 | 345 |
| 0 | 0 | 52 | 46 | 12 | 1010 | 1428 |
| 0 | 1 | 56 | 41 | 12 | 438 | 582 |
| 0 | 37 | 38 | 23 | 4 | 232 | 232 |
| 0 | 10 | 38 | 52 | 4 | 1119 | 1762 |
| 0 | 6 | 18 | 75 | 10 | 878 | 1400 |
| 0 | 11 | 39 | 48 | 4 | 734 | 943 |
| 1 | 0 | 57 | 42 | 12 | 1230 | 1733 |
| 8 | 26 | 33 | 32 | 4 | 1417 | 3386 |
| 0 | 0 | 81 | 19 | 8 | 1319 | 2842 |
| 0 | 1 | 69 | 29 | 4 | 334 | 462 |
| 0 | 6 | 18 | 75 | 8 | 851 | 1461 |
| 3 | 33 | 30 | 33 | 4 | 708 | 1200 |
| 1 | 0 | 70 | 29 | 12 | 1005 | 1994 |
| 0 | 10 | 38 | 52 | 12 | 1056 | 1407 |
| 1 | 0 | 70 | 29 | 8 | 1069 | 2282 |
| 0 | 18 | 42 | 39 | 4 | 136 | 234 |
| 0 | 0 | 52 | 46 | 2 | 206 | 258 |
| 0 | 0 | 68 | 31 | 2 | 222 | 215 |
| 0 | 1 | 69 | 29 | 12 | 334 | 427 |

**Table 3. Statistical analysis of collected database.**

| Description | G | S | M | C | L | UCS7 | UCS28 |
|---|---|---|---|---|---|---|---|
| | % | % | % | % | % | kPa | kPa |
| Training set | | | | | | | |
| Min. | 0.00 | 0.00 | 13.00 | 12.00 | 2.00 | 127 | 91 |
| Max. | 10.00 | 45.00 | 81.00 | 75.00 | 14.00 | 2335 | 4219 |
| Avg. | 1.51 | 14.62 | 42.51 | 40.34 | 7.34 | 880 | 1489 |
| SD | 3.14 | 14.72 | 17.71 | 15.28 | 3.52 | 495 | 978 |
| VAR | 2.08 | 1.01 | 0.42 | 0.38 | 0.48 | 0.56 | 0.66 |
| Validation set | | | | | | | |
| Min. | 0.08 | 0.06 | 0.00 | 1.79 | 139.00 | 0.79 | |
| Max. | 0.93 | 0.92 | 0.05 | 2.10 | 205.00 | 69.70 | |
| Avg. | 0.30 | 0.69 | 0.00 | 1.89 | 157.00 | 24.81 | |
| SD | 0.27 | 0.28 | 0.01 | 0.09 | 22.96 | 24.44 | |
| VAR | 0.91 | 0.41 | 2.60 | 0.05 | 0.15 | 0.99 | |

characteristics and the Pearson correlation matrix respectively. Lastly, Fig 2 shows the histograms for both inputs and outputs.

## 3. Research program

Three different Artificial Intelligent (AI) techniques and one symbolic machine learning trained technique were used to predict the unconfined compressive strengths after 7 and 28 days (UCS7, UCS28) of open-air curing of the lime reconstituted soil using the collected, sorted and organized database. These AI techniques are the Genetic Programming (GP), three models trained differently of Artificial Neural Network (ANN) and polynomial regression optimized using genetic algorithm which is known as the Evolutionary Polynomial Regression (EPR) [49–51]. The symbolic machine learning trained technique is known as the response surface methodology due to its simple interface of proposing a closed-form equation that allows researchers to apply its model automatically and manually [52–59]. All the four (4) developed models were deployed to predict (UCS7, UCS28) in (kPa) using the soil grading proportions from Gravel, Sand, Silt, and Clay and additive; Lime contents (G, S, M, C & L). Each of the four developed models was based on a different approach: an evolutionary approach for GP, mimicking biological neurons for ANN, an optimized mathematical regression technique for EPR and symbolic regression interface algorithm for the RSM. However, all the models were assessed for prediction accuracy based on the Sum of Squared Errors (SSE) and the determination coefficient known as the R-squared value ($R^2$). In the data partitioning, the k-fold cross-validation has been applied to solve problems of under-fitting due to the size

**Table 4. Pearson correlation matrix of the dataset.**

| | G | S | M | C | L | UCS7 | UCS28 |
|---|---|---|---|---|---|---|---|
| G | 1.00 | | | | | | |
| S | 0.35 | 1.00 | | | | | |
| M | -0.23 | -0.61 | 1.00 | | | | |
| C | -0.26 | -0.33 | -0.53 | 1.00 | | | |
| L | -0.03 | -0.03 | -0.08 | 0.14 | 1.00 | | |
| UCS7 | 0.28 | 0.22 | -0.26 | 0.05 | 0.30 | 1.00 | |
| UCS28 | 0.31 | 0.19 | -0.18 | -0.02 | 0.32 | 0.95 | 1.00 |

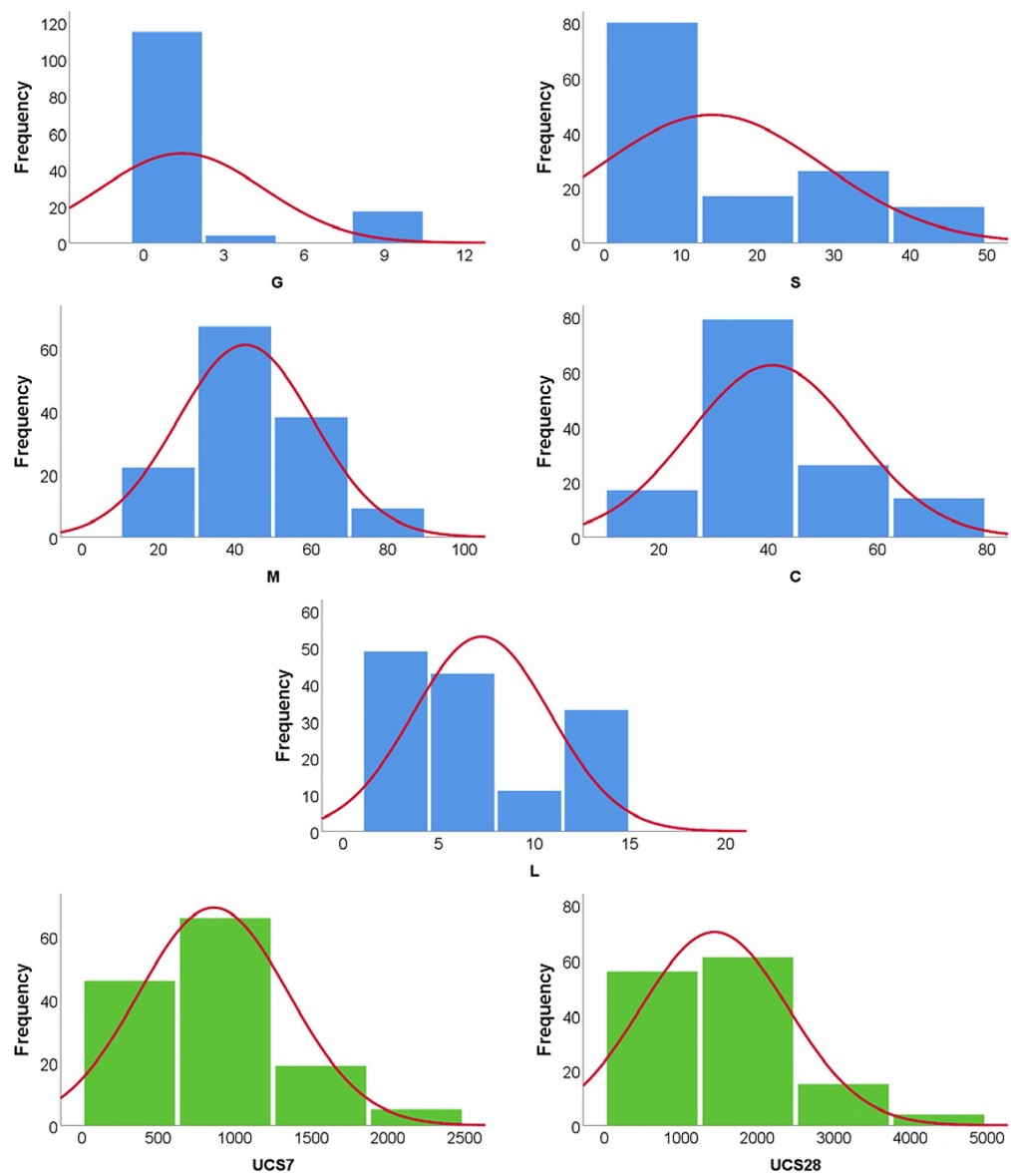

**Fig 2. Distribution histograms for inputs (in blue) and outputs (in green).**

of the data points [58,59]. The accuracies of the developed models were evaluated by comparing the SSE between predicted and calculated shear strength parameter values based on the results of each model.

### 3.1 Genetic Programming (GP)

Genetic programming (GP), the framework of which is illustrated in Fig 3 is a type of evolutionary algorithm that is used to automatically generate computer programs to solve problems or perform tasks [52]. It is a machine learning technique that draws inspiration from the process of natural selection and genetic evolution [53]. The basic idea behind genetic programming is to create a population of candidate computer programs, represented as trees or graphs,

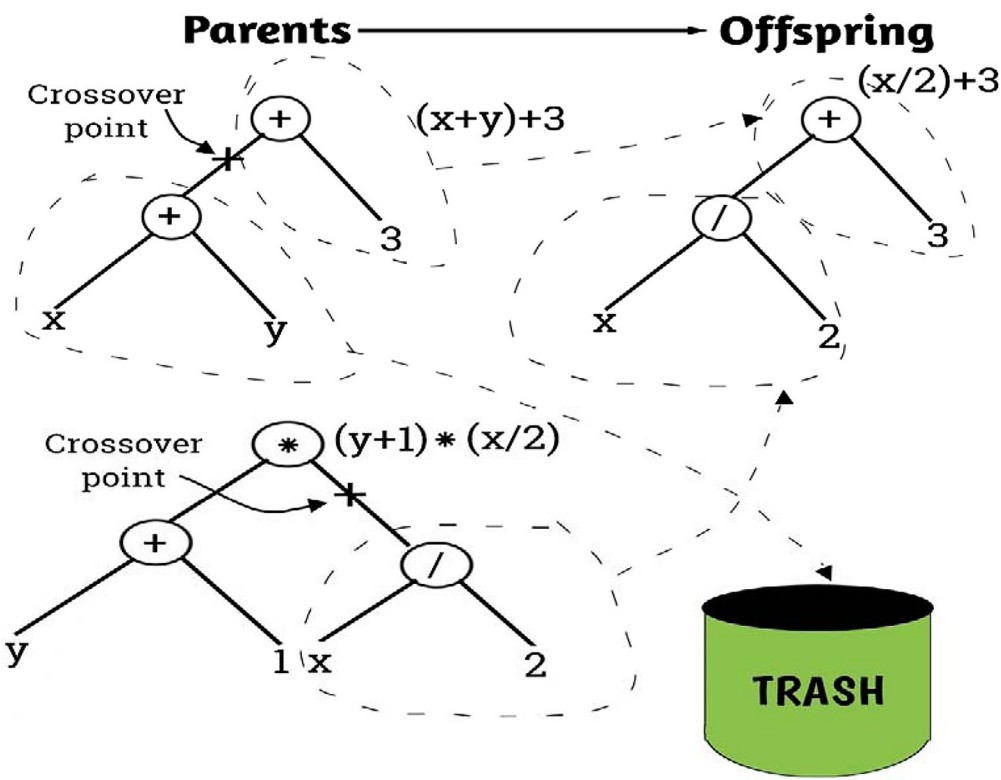

**Fig 3. Typical illustrative framework for the genetic programming model.**

and then use evolutionary principles such as selection, crossover, and mutation to evolve and improve these programs over successive generations until a desired level of performance is achieved. Representation: In genetic programming, computer programs are typically represented as trees or graphs, with nodes representing operations or functions, and edges representing the flow of data or control between operations. Initialization: The process starts by creating an initial population of random programs [54]. Each program is evaluated based on its ability to solve the given problem or perform the desired task. Selection: Programs from the current population are selected for reproduction based on their fitness, which is a measure of how well they perform the task. This is typically done using a fitness function that quantifies the performance of each program. Crossover: Selected programs are combined through crossover, a process that mimics genetic recombination in nature [55]. During crossover, sub-trees or sub-graphs from two parent programs are exchanged to create new offspring programs. Mutation: Random changes are introduced to the offspring programs through mutation, simulating genetic variation. This helps in exploring new regions of the search space and preventing premature convergence to suboptimal solutions. Evaluation: The newly created programs are evaluated using the fitness function to determine their performance on the given task. Termination: The evolutionary process continues for a certain number of generations or until a termination condition is met, such as reaching a satisfactory level of performance or running out of computational resources [54]. Genetic programming has been successfully applied to a wide range of problems, including symbolic regression, automatic program synthesis, control system design, and pattern recognition [52–56]. It is a powerful approach for automatically discovering solutions to complex problems without the need for human-designed algorithms.

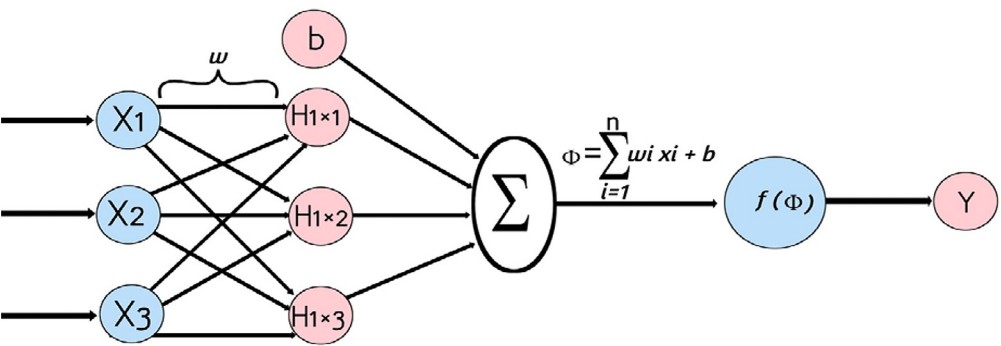

**Fig 4. Typical illustrative framework for the ANN.**

## 3.2 Artificial Neural Network (ANN)

An artificial neural network (ANN), the typical framework if which is illustrated in Fig 4 is a computational model inspired by the structure and function of biological neural networks, such as the human brain [49]. It is a powerful machine learning technique used for solving complex problems such as pattern recognition, classification, regression, and more [50]. Structure: ANN consists of interconnected nodes, called neurons or units, organized in layers. Typically, there are three types of layers: input layer, hidden layers, and output layer [51]. The connections between neurons are associated with weights that are adjusted during the training process. Learning: ANNs learn from data through a process called training. During training, the network is presented with input data, and the weights of the connections are adjusted based on the network's output and the expected output [50]. This process is often performed using optimization algorithms such as gradient descent and its variants. Activation Function: Each neuron in a neural network typically applies an activation function to the weighted sum of its inputs. Common activation functions include sigmoid, tanh, ReLU (Rectified Linear Unit), and their variants [49]. Types of Neural Networks: There are various architectures of neural networks, such as feedforward neural networks (the most basic type), convolutional neural networks (CNNs) for image processing, recurrent neural networks (RNNs) for sequential data, and more advanced architectures like deep neural networks (DNNs) and generative adversarial networks (GANs) [50]. Applications: ANNs are used in a wide range of applications, including image and speech recognition, natural language processing, recommendation systems, financial forecasting, medical diagnosis, and many other fields where complex pattern recognition and prediction tasks are required [51]. Challenges: Training a neural network can be computationally intensive and requires a large amount of labeled data. Overfitting, where a model performs well on training data but poorly on unseen data, is a common challenge that needs to be addressed through techniques like regularization and cross-validation. Overall, artificial neural networks have proven to be highly effective in solving complex problems, and their capabilities continue to expand with ongoing research and advancements in the field of deep learning.

## 3.3 Evolutionary Polynomial Regression (EPR)

Evolutionary Polynomial Regression (EPR) is a non-linear regression technique that uses a genetic programming approach to evolve mathematical models [49]. It was proposed by Dr. Nordin Zakaria in the early 1990s. EPR combines the concepts of genetic algorithms and polynomial regression to automatically evolve a mathematical model that best fits a given dataset. The algorithm starts with a population of random mathematical expressions (polynomials)

and uses genetic operators such as selection, crossover, and mutation to evolve these expressions over several generations [51]. During the evolutionary process, the algorithm evaluates the fitness of each mathematical expression by comparing its performance in fitting the dataset [50]. The fittest expressions are then selected to produce offspring through genetic operators, and this process is repeated for multiple generations until a satisfactory mathematical model is obtained [49]. EPR has been used in various fields, including engineering, economics, and environmental science, to model complex relationships between input variables and output data [50]. It has the advantage of being able to automatically discover the structure of the mathematical model, making it particularly useful when the underlying relationship between variables is not well understood or when traditional regression techniques may not be suitable [49]. However, it's worth noting that EPR, like other evolutionary algorithms, can be computationally intensive and may require careful parameter tuning to achieve optimal results. Additionally, the interpretability of the evolved models can be a challenge, as the resulting mathematical expressions may be complex and difficult to understand.

## 3.4 Models' evaluation indices

Machine learning model performance evaluation is a critical step in the development and deployment of machine learning models. It involves assessing how well a trained model performs on new, unseen data. There are several methods for evaluating the performance of machine learning models, and the choice of method depends on the type of problem and the nature of the data [49]. Here are some commonly used techniques for evaluating machine learning model performance: Train/Test Split: The simplest method for evaluating model performance is to split the available data into a training set and a testing set [50]. The model is trained on the training set and then evaluated on the testing set to assess its performance on unseen data. Cross-Validation: Cross-validation is a technique used to assess how well a model generalizes to new data [51]. It involves splitting the data into multiple subsets, training the model on a combination of these subsets, and then evaluating it on the remaining subset. This process is repeated multiple times, and the results are averaged to obtain a more reliable estimate of the model's performance. Performance Metrics: Various performance metrics can be used to evaluate the performance of machine learning models, depending on the nature of the problem [49]. Common metrics include accuracy, precision, recall, F1 score, area under the receiver operating characteristic (ROC) curve (AUC-ROC), and mean squared error (MSE), among others. Confusion Matrix: For classification problems, a confusion matrix can be used to visualize the performance of a model by showing the number of true positive, false positive, true negative, and false negative predictions [51]. ROC Curve and Precision-Recall Curve: These curves are used to evaluate the performance of binary classification models and visualize the trade-off between true positive rate and false positive rate, or precision and recall, respectively [50]. Bias-Variance Trade-off: Understanding the bias-variance trade-off is crucial for evaluating model performance. A model with high bias may underfit the data, while a model with high variance may overfit the data. Balancing bias and variance are important for creating a model that generalizes well to new data [49]. Hyperparameter Tuning: Evaluating model performance often involves hyperparameter tuning, which involves adjusting the settings of a model to optimize its performance. Techniques such as grid search and random search can be used to find the best hyperparameters for a given model. It's important to note that the choice of evaluation method and performance metric depends on the specific machine learning problem at hand, and there is no one-size-fits-all approach. Additionally, it's important to consider the implications of the chosen evaluation method on the overall goals of the machine learning project [50]. Meanwhile, data partitioning and k-fold cross-validation are both techniques

used in machine learning for model evaluation and selection. Each approach has its own effects and considerations, and they can be complementary in addressing various challenges in model training and assessment. In data partitioning, the dataset is typically divided into two subsets: a training set and a test set. The training set is used to train the model, while the test set is used to evaluate the model's performance on unseen data [51]. Commonly, data partitioning involves a 70/30 or 80/20 split, where the larger portion of the data is used for training and the smaller portion for testing. This approach is simple and easy to implement. K-fold cross-validation involves dividing the dataset into K subsets (folds) [50]. The model is trained and evaluated K times, each time using a different fold as the validation set and the remaining folds as the training set. Performance metrics are averaged across the K iterations to obtain a final performance estimate. K-fold cross-validation provides a more comprehensive assessment of the model's performance, as it uses the entire dataset for both training and validation. Data partitioning can lead to variability in model performance estimates, especially when the test set is small [49]. The performance of the model may heavily depend on which data points end up in the test set. K-fold cross-validation provides a more reliable estimate of model performance, as it averages performance across multiple validation sets, reducing the impact of variability in the test set [50]. With data partitioning, a portion of the data is reserved solely for testing, which means that less data is available for model training K-fold cross-validation makes more efficient use of the available data, as each data point is used for both training and validation at some point during the K iterations [50]. Data partitioning may lead to a risk of overfitting to the test set, especially when the test set is relatively small [49]. K-fold cross-validation can help mitigate the risk of overfitting to a single test set, as the model is evaluated on multiple validation sets, providing a more robust assessment of its generalization performance [49]. In practice, both techniques can be used in combination. For example, a dataset can be partitioned into a training set and a holdout test set, and K-fold cross-validation can be applied to the training set for model selection and hyperparameter tuning [49]. This combined approach allows for robust model evaluation while still reserving a separate test set for final model assessment. K-fold cross-validation is a powerful technique for assessing and mitigating overfitting in machine learning models. Overfitting occurs when a model learns to perform well on the training data but does not generalize well to unseen data [51]. K-fold cross-validation helps to address overfitting in the following ways: By using K-fold cross-validation, the model is evaluated multiple times on different subsets of the data. This process allows for a more comprehensive understanding of how well the model generalizes to unseen data [50]. If a model performs well across all K folds, it is an indication that the model is less likely to be overfitting [49]. When a model is trained and evaluated on a single train-test split, the performance estimate can be highly dependent on which data points end up in the training set and which end up in the test set. This can lead to high variance in the performance estimate. K-fold cross-validation helps to reduce this variance by averaging performance across K different validation sets, providing a more stable estimate of model performance [49]. In each fold of K-fold cross-validation, every data point is used for both training and validation. This ensures that all data points contribute to the evaluation of the model, which can help in identifying overfitting tendencies that may not be apparent when using a single train-test split [50]. K-fold cross-validation can be used to compare the performance of multiple models and select the one that generalizes best to unseen data. This can help in choosing a model that is less prone to overfitting [50]. When tuning hyperparameters of a model, K-fold cross-validation can be used to find the optimal settings while guarding against overfitting to the validation set. Overall, K-fold cross-validation is a valuable tool for assessing and addressing overfitting in machine learning models [50]. It provides a more robust evaluation of model performance and helps in selecting models that are more likely to generalize well to new data.

**3.4.1 Sum of Squared Errors (SSE).**   The sum of squared errors (SSE), also known as the residual sum of squares (RSS), is a commonly used metric for evaluating the performance of a regression model. It is a measure of the discrepancy between the observed values and the values predicted by the model [59]. In the context of linear regression, the SSE is calculated by taking the difference between each observed target value (i.e., the actual value in the dataset) and the corresponding predicted value (i.e., the value predicted by the regression model), squaring each difference, and then summing up all these squared differences [58,59]. Mathematically, the SSE for a regression model with n data points can be expressed as:

$$SSE = \Sigma(y_i - \overline{y})^2 \tag{1}$$

Where: $y_i$ represents the observed value of the target variable for the i-th data point, $\overline{y}$ represents the predicted value of the target variable for the i-th data point, and the summation is taken over all n data points. The SSE is a measure of the variability of the data that is not explained by the regression model [49]. A lower SSE indicates a better fit of the model to the data, as it means that the model's predictions are closer to the actual observed values. The SSE is often used in the context of ordinary least squares (OLS) regression, where the goal is to minimize the SSE to find the best-fitting line or hyperplane for the given data [50]. Minimizing the SSE is equivalent to finding the parameters of the regression model that provide the best fit to the data in terms of minimizing the squared differences between the observed and predicted values [59]. While the SSE is a useful metric for evaluating the performance of regression models, it is important to consider other metrics as well, such as R-squared and others, to gain a comprehensive understanding of the model's performance.

**3.4.2 R-squared value.**   R-squared, often denoted as $R^2$, is a statistical measure that represents the proportion of the variance in the dependent variable that is predictable from the independent variable(s) in a regression model [59]. In the context of linear regression, it is a measure of how well the independent variables explain the variability of the dependent variable. The $R^2$ value is calculated as the ratio of the explained sum of squares (ESS) to the total sum of squares (TSS), and is defined as:

$$R\hat{2} = 1 - \frac{SSE}{TSS} \tag{2}$$

Where: SSE denotes the sum of squared errors (also known as residual sum of squares), which measures the discrepancy between the observed values and the values predicted by the model. TSS represents the total sum of squares, which measures the total variance in the dependent variable. Alternatively, the $R^2$ value can also be calculated as the squared correlation coefficient between the observed and predicted values of the dependent variable [58]. This interpretation underscores the notion that $R^2$ measures the proportion of the variance in the dependent variable that is explained by the independent variables in the model. The $R^2$ value ranges from 0 to 1, with: $R^2 = 0$ indicating that the independent variables do not explain any of the variability of the dependent variable. $R^2 = 1$ indicating that the independent variables explain all of the variability of the dependent variable [59]. Interpretation of $R^2$: A higher $R^2$ value indicates that a larger proportion of the variance in the dependent variable is explained by the independent variables, suggesting a better fit of the model to the data. A lower $R^2$ value indicates that the independent variables provide little explanatory power for the dependent variable [57]. It's important to note that while $R^2$ is a useful measure for assessing the goodness of fit of a regression model, it should be used in conjunction with other metrics, such as adjusted $R^2$, mean squared error (MSE), and others, to gain a comprehensive understanding of the model's performance and to avoid potential pitfalls associated with overfitting or underfitting.

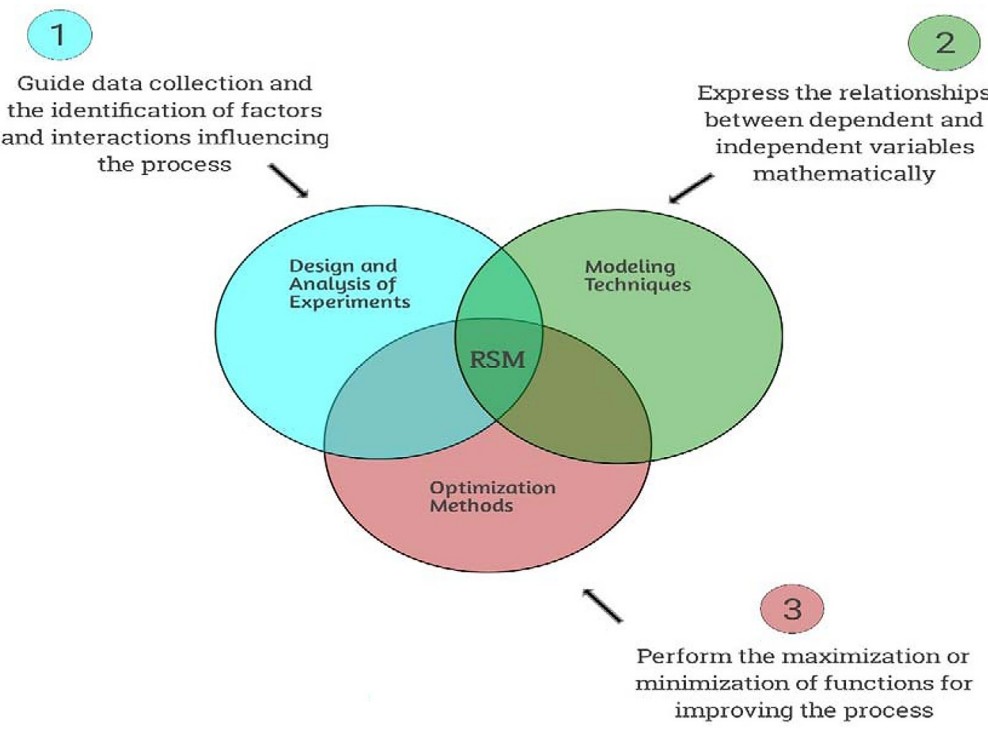

**Fig 5. Typical framework for the RSM model.**

## 3.5 Response Surface Methodology (RSM)

Response Surface Methodology (RSM) the framework of which is shown in Fig 5 is a collection of mathematical and statistical techniques used for designing experiments, building empirical models, and finding the optimal conditions for a process [52]. It is commonly used in scientific and engineering fields, particularly in areas such as chemical engineering, manufacturing, and product development. Modeling and optimizing a process: RSM is used to develop mathematical models that represent the relationship between input variables (factors) and the output response of interest [53]. These models are then used to optimize process conditions to achieve desired outcomes. Experiment design: RSM involves the design of experiments to efficiently explore the relationships between multiple input variables and the response variable [54]. This typically involves conducting a series of experiments with carefully chosen factor settings to collect data for model building. Finding optimal process conditions: Once the mathematical models are developed, RSM techniques are used to identify the optimal or near-optimal settings of the input variables that lead to the desired response [55]. Factorial designs: These are experimental designs in which all possible combinations of factor levels are studied. They are used to identify the main effects and interactions of the factors on the response [56]. Central composite designs: These designs involve a combination of factorial points and center points, and are used to fit a second-order polynomial model. They are effective for estimating curvature and interaction effects [57]. Box-Behnken designs: These designs are used to fit a second-order model without needing to study all possible combinations of factor levels. They are particularly useful when the number of factors is moderate. Analysis of variance (ANOVA): ANOVA is used to analyze the significance of the factors and their interactions on the response variable [54]. Response surface optimization: Optimization techniques are used to find the optimal settings of the input variables that lead to the best or desired response [56]. Overall,

Response Surface Methodology provides a systematic and efficient approach for understanding and optimizing complex processes by using empirical models and statistical techniques.

## 4. Results and discussion

### 4.1 GP model technique

The developed GP model has four levels of complexity. The population size, survivor size and number of generations were 100 000, 30 000 and 100 respectively. Eqs 3 and 4 presented the output formula for (UCS7and UCS28) respectively. The average errors % of total dataset are (0.24, 0.34), while the $R^2$ values are 0.777 and 0.650 in order. The proposed closed-form equation suggests that the proportion of silt and clay are more decisive in the behavior of the compressive strength of the lime reconstituted soil at the 7 days curing stage.

$$\text{UCS7} = \frac{2(M+C)}{G+S-M} - \left( \frac{(C-2M)^2}{C-S-G} + 2G(M+C) + 773 \right) \left( \frac{2.77 + C - 2(S+G)}{1.77 + C - 2(S+G)} \right)$$
$$+ \frac{2(M+C)(3C-2M)L - 8(M+C)^2}{2L(M+C)(C-M)^{(C-2M)} + M.L} + \frac{2S + 2G - 4M - 5C}{L} + (3C - 2M) \quad (3)$$

$$\text{UCS28} = \frac{1600(L-2)}{L} + \frac{(2GS+1)(2M-2L) - 2GS(S+G)}{2S-40} + \frac{2(2S-40)^2}{M-L}$$
$$+ \left( \frac{3L(G-S+3M-C-2L) - 1600(L-2)}{L} + \frac{2GS(G-S) + (4GS+1)(M-L)}{2S-40} \right)$$
$$\times \frac{1}{2(S-M)^s} + 4(GS+C-M) \quad (4)$$

### 4.2 ANN model technique

A predictive model was developed using ANN technique to predict both UCS7 and UCS28 values. It used normalization method (-1.0 to 1.0), activation function (Hyper Tan) and "Back propagation" (BP) training algorithm. The ANN developed model weight matrix is presented in Table 5. The used network layout is illustrated in Fig 6 while the weight matrix of the model is showed in Table 3. The average errors in percentage for the total dataset were found to be 12% and 15%, with corresponding $R^2$ values of 0.952 and 0.947, respectively. The relative importance values for each input parameter are illustrated in Fig 7, which indicated that all factors have almost the same importance for both UCS7 and UCS28 except the Lime content (L)

**Table 5. Weights matrix for the developed ANN.**

|          | H(1:1) | H(1:2) | H(1:3) | H(1:4) | H(1:5) | H(1:6) | H(1:7) | H(1:8) |        |
|----------|--------|--------|--------|--------|--------|--------|--------|--------|--------|
| (Bias)   | -7.28  | -6.55  | -1.44  | 1.70   | 0.44   | -8.15  | -8.91  | -4.32  |        |
| G        | 5.93   | 12.85  | 2.66   | -0.18  | 0.61   | 3.19   | 3.12   | -1.73  |        |
| S        | 1.31   | -4.39  | -7.32  | -0.90  | -0.20  | -3.18  | -4.94  | 5.32   |        |
| M        | 0.42   | 10.25  | -8.99  | 2.76   | -2.24  | -2.47  | -4.29  | -3.90  |        |
| C        | -11.69 | 8.82   | -7.79  | 3.39   | 1.63   | 2.59   | 1.38   | -4.89  |        |
| L        | -3.84  | 0.46   | -2.71  | 0.36   | -0.48  | -2.12  | -2.21  | -0.21  |        |
|          | H(1:1) | H(1:2) | H(1:3) | H(1:4) | H(1:5) | H(1:6) | H(1:7) | H(1:8) | (Bias) |
| UCS7     | 0.26   | 0.72   | -0.53  | 3.87   | 0.76   | -2.99  | 2.88   | 3.91   | -0.53  |
| UCS28    | 0.31   | 0.95   | -0.74  | 3.63   | 0.89   | -4.60  | 4.48   | 3.73   | -0.60  |

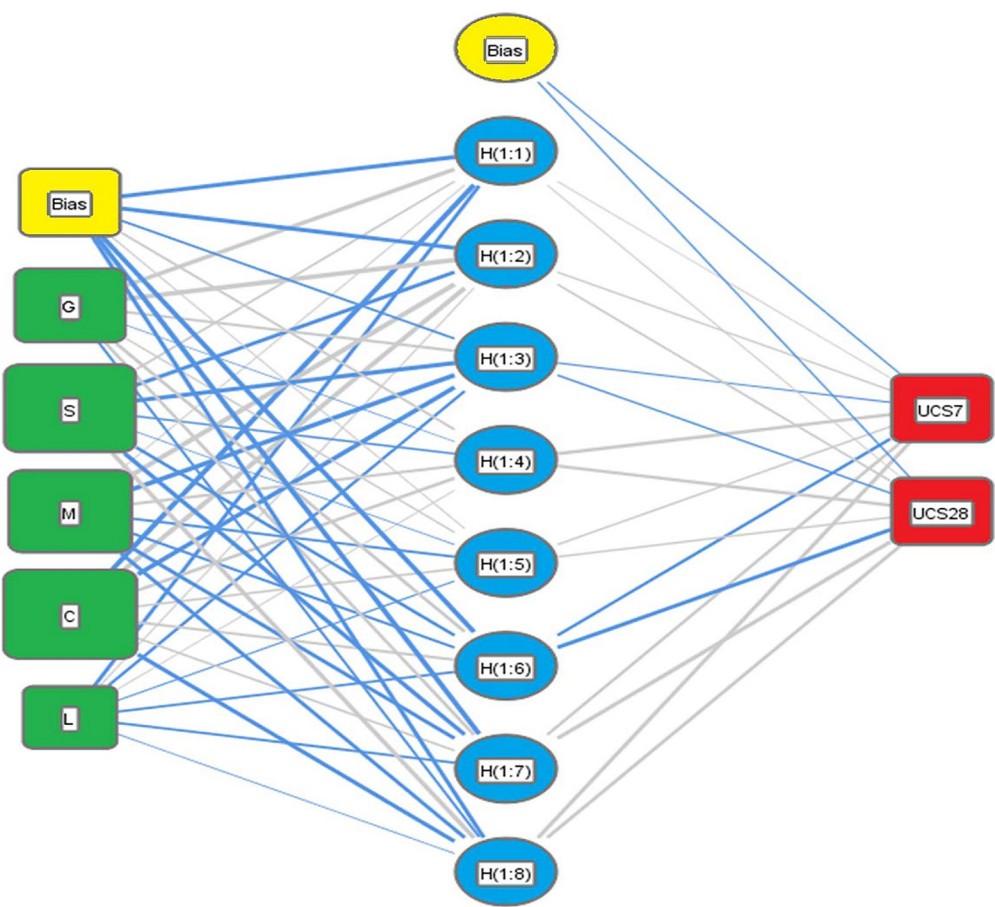

**Fig 6. Layout for the developed ANN models.**

which has less influence. It further shows that sand (S) and clay (C) are the most impactful on the strength response of the lime-reconstituted graded soil, which corroborates with previous reports [51, 65]. This shows that the finer materials perform better with the strength improvement of landfill liners and subgrades under lime treatment [51].

### 4.3 EPR model technique

Finally, the developed EPR model was limited to $6^{th}$ level polynomial, for 5 inputs, there are 462 possible terms (252+126+56+21+6+1 = 462) as follows:

$$\sum_{n=1}^{n=5} \sum_{m=1}^{m=5} \sum_{l=1}^{l=5} \sum_{k=1}^{k=5} \sum_{j=1}^{j=5} \sum_{i=1}^{i=5} X_n.X_m.X_l.X_k.X_j.X_i$$
$$+ \sum_{m=1}^{m=5} \sum_{l=1}^{l=5} \sum_{k=1}^{k=5} \sum_{j=1}^{j=5} \sum_{i=1}^{i=5} X_m.X_l.X_k.X_j.X_i$$
$$+ \sum_{l=1}^{l=5} \sum_{k=1}^{k=5} \sum_{j=1}^{j=5} \sum_{i=1}^{i=5} X_l.X_k.X_j.X_i + \sum_{k=1}^{k=5} \sum_{j=1}^{j=5} \sum_{i=1}^{i=5} X_k.X_j.X_i$$
$$+ \sum_{j=1}^{j=5} \sum_{i=1}^{i=5} X_j.X_i + \sum_{i=1}^{i=5} X_i + C \tag{5}$$

GA technique was applied on these 462 terms to select the most effective 47 terms to predict the values of UCS7 and 50 terms to predict UCS28 values. The average error percentages and $R^2$ values for UCS7 and UCS28 were found to be 12% and 18% and 0.955 and 0.923,

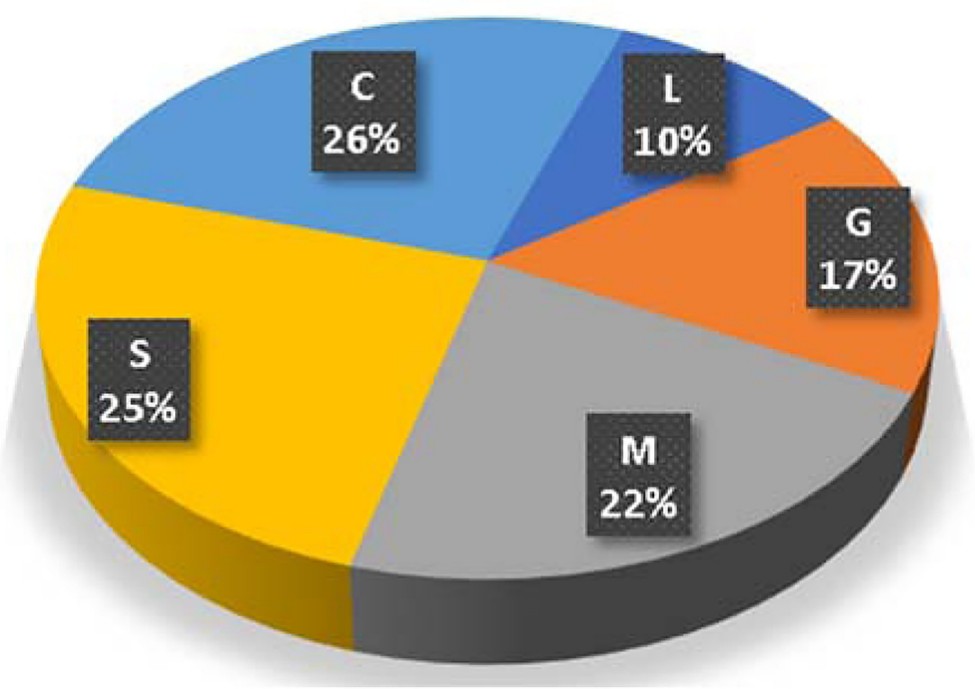

**Fig 7. Relative importance of input parameters.**

respectively.

$$
\begin{aligned}
UCS7 =\ & \frac{GSC^3}{1275} - \frac{SMCL}{252} + \frac{SMC^2}{2140L} - \frac{758530\ SM}{C^2} + \frac{1052SM^2}{C} + \frac{SCM^3}{15.7} - \frac{SM^3C^2}{3220} + \frac{1.83\ SM^3}{C} + \frac{130\ SM^3}{C^2} \\
& + \frac{700500\ S}{M} - \frac{4.07\ SCL}{M} + \frac{SCL^2}{23.5M} - \frac{SC^3}{61.1\ ML} + \frac{4.22 \times 10^7\ S}{MC} + \frac{S^2MC^2L}{180400} + \frac{S^2ML}{5.13\ C^2} + \frac{21680\ S^2}{M} \\
& - \frac{123740000S^2}{CM^3} - \frac{MCL}{28.8} + \frac{MCL^2}{1812} - \frac{MC^2}{142.5L} + \frac{MC^3}{228.3} - \frac{60256\ M}{C} - \frac{230.5\ M}{CL} - \frac{CM^3}{63.4} \\
& + \frac{6.674 \times 10^6}{M} - \frac{9.5 \times 10^7}{MC} + \frac{3.855 \times 10^{12}}{M^2C^3} + \frac{718.6\ CL^2}{M^3} - \frac{33910\ C^2}{M^3} - \frac{96\ LC^2}{M^3} - \frac{7.664 \times 10^{13}}{M^3C^3} \\
& - \frac{9.447 \times 10^7}{M^3L^3} + \frac{C^2L^2}{79.68} - \frac{C^3L^2}{6780} + \frac{149.7\ L^2}{C} + \frac{68.4\ L^3}{C^2} - 0.75\ L^3 - 75SM^2 - 3.07\ SCM^2 \\
& - 226568\ S + 6890\ SM + 85.8S\ MC - 1.14\ SM^3 - 584\ S^2 + 9595M - 447152
\end{aligned}
\tag{6}
$$

$$
\begin{aligned}
UCS28 =\ & \frac{1828.4\ SG^3}{MC} - \frac{GSLM^3}{10575} - \frac{SCLG^3}{11760} - \frac{MCLG^3}{2040} - \frac{4.55MG^3}{CL} + \frac{SMCL}{122.4} - \frac{SMCL^2}{2940} - \frac{SMC^2}{3725L} \\
& - \frac{195310\ SM}{C^3} + \frac{CLSM^2}{5060} + \frac{CSM^3}{91.9} - \frac{CLSM^3}{132.6} + \frac{SM^3C^2}{53.4} - \frac{480.7\ SC^2}{M} + \frac{2.98SC^3}{M} + \frac{978514\ S}{M^2} \\
& - \frac{S^2M^2C}{442.8} - \frac{S^2CL^2}{205.6M} - \frac{5.56\ S^2C^2}{M} + \frac{0.167\ S^2C^3}{M} + \frac{S^2CL^2}{10210} - \frac{MCL^2}{49.8} + \frac{MCL^3}{1890} - \frac{70.3\ M}{CL} \\
& + \frac{1104.7\ ML}{C^2} - \frac{M^2C^3}{6675} + \frac{71\ M^2}{C} - \frac{87M^2}{CL^3} - \frac{21042\ M^2}{C^3} + \frac{33583\ C}{LM2} + \frac{136.2\ C^3}{M^2} + \frac{8963\ L^3}{CM^2} \\
& - \frac{4.165 \times 10^8}{M^3L^3} + \frac{138233}{CL^2} + \frac{1843}{L^2} - 1.76CM^2 + 3806\ M + 159.5\ MC + 0.27\ MCL - 169.3\ S^2 \\
& - 0.765\ SM^3 + 107.1\ SM^2 - 1.73\ CSM^2 + 64422\ S - 4294\ SM + 54.4\ SMC + 0.358\ MLG^2 \\
& - 8.06\ MG^3 + 0.183\ MCG^3 - 331075
\end{aligned}
\tag{7}
$$

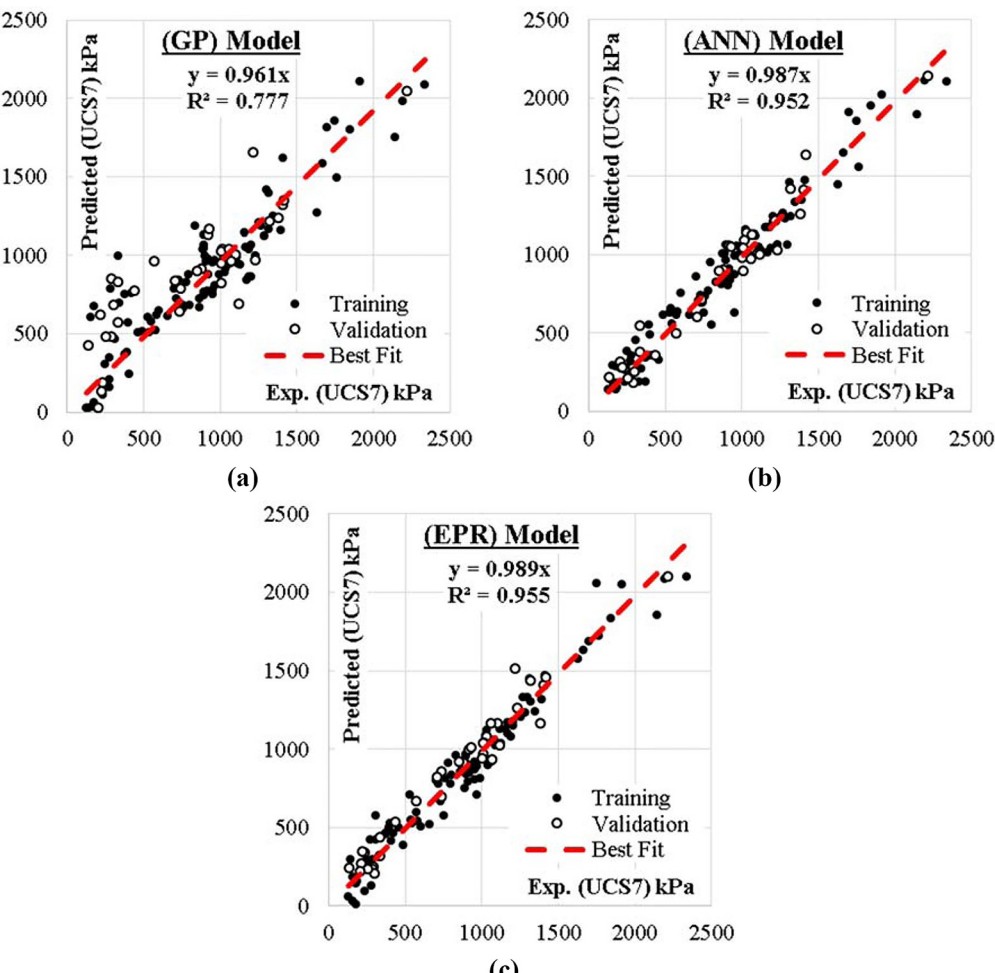

**Fig 8. Relation between predicted and calculated (UCS7) values using the developed models.**

The relations between calculated and predicted values are shown in Figs 8 and 9 and the results of all the developed models are summarized in Table 6. In Fig 10, the accuracies of the developed models are compared by using Taylor charts for the UCS7 and UCS28.The data entries are more concentrated around 500 to 1500 kPa for the UCS7 and around 900 to 2700 kPa for the UCS28 for the training and validation entries of the three models as shown in Figs 8 and 9. The best fit for the ANN with the line equation of y = 0.987x and y = 980x for the 7 days cured unconfined compressive strength (UCS7) and the 28 days cured unconfined compressive strength (UCS28), respectively show the best efficient model execution even though the ANNs models did not produce closed-form equations, which may allow for a manual application of the superior model. The Taylor diagram agrees with the performance model shown in Figs 8 and 9, which shows the ANN and EPR in the 0.95–0.99 segment of the accuracy diagram in Fig 10. Finally, the closed-form equations especially the EPR model are applicable in the design and construction of landfill liners and subgrade to determine the optimized compressive strength of the compacted earth layer as the foundation course for a flexible pavement at 250 kN/m$^2$ and compacted earth liner at 200 kN/m$^2$ strength surfaces to determine the allowable strength for sustainable liner courses for a lime reconstituted gravel-sand-silt-clay (G-S-M-C) graded soil, however this is supported by previous research works [1,3,4,65].

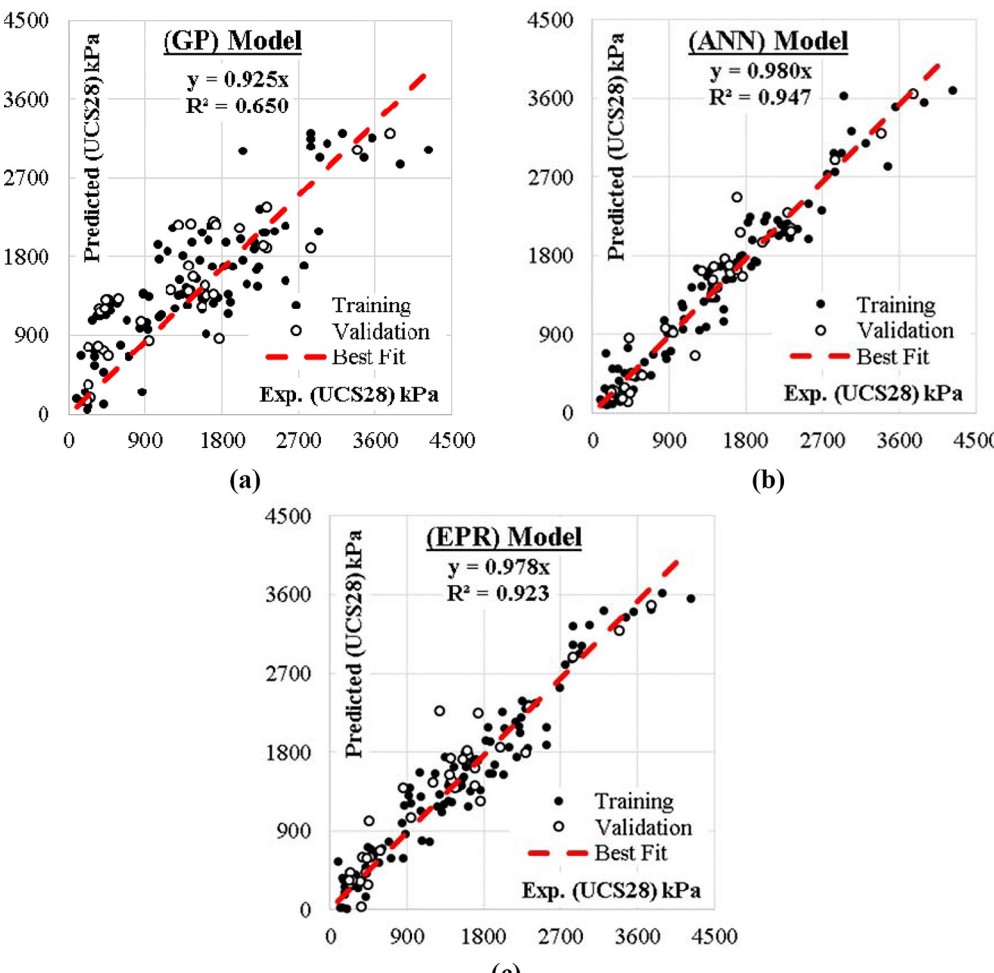

**Fig 9. Relation between predicted and calculated (UCS28) values using the developed models.**

The ANN model can only be applied within the intelligent interface because it didn't produce a closed-form equation with which a manual application is executed.

## 4.4 RSM model analysis

The Predicted $R^2$ for the UCS 7 of 0.9879 is in reasonable agreement with the Adjusted $R^2$ of 0.8805; i. e. the difference is less than 0.2. Adeq precision measures the signal to noise ratio. A ratio greater than 4 is desirable. Your ratio of 14.169 indicates an adequate signal. This model

**Table 6. Accuracies of developed models.**

| Item | Technique | Model | SSE | Avg. Error (%) | $R^2$ |
|------|-----------|-------|-----|----------------|-------|
| UCS7 | GP | Eq 3 | 6,037,178 | 24 | 0.777 |
| | ANN | Fig 3, Table 3 | 1,496,464 | 12 | 0.952 |
| | EPR | Eq 6 | 1,403,965 | 12 | 0.955 |
| UCS28 | GP | Eq 4 | 33,340,605 | 34 | 0.650 |
| | ANN | Fig 3, Table 4 | 6,679,788 | 15 | 0.947 |
| | EPR | Eq 7 | 8,864,671 | 18 | 0.923 |

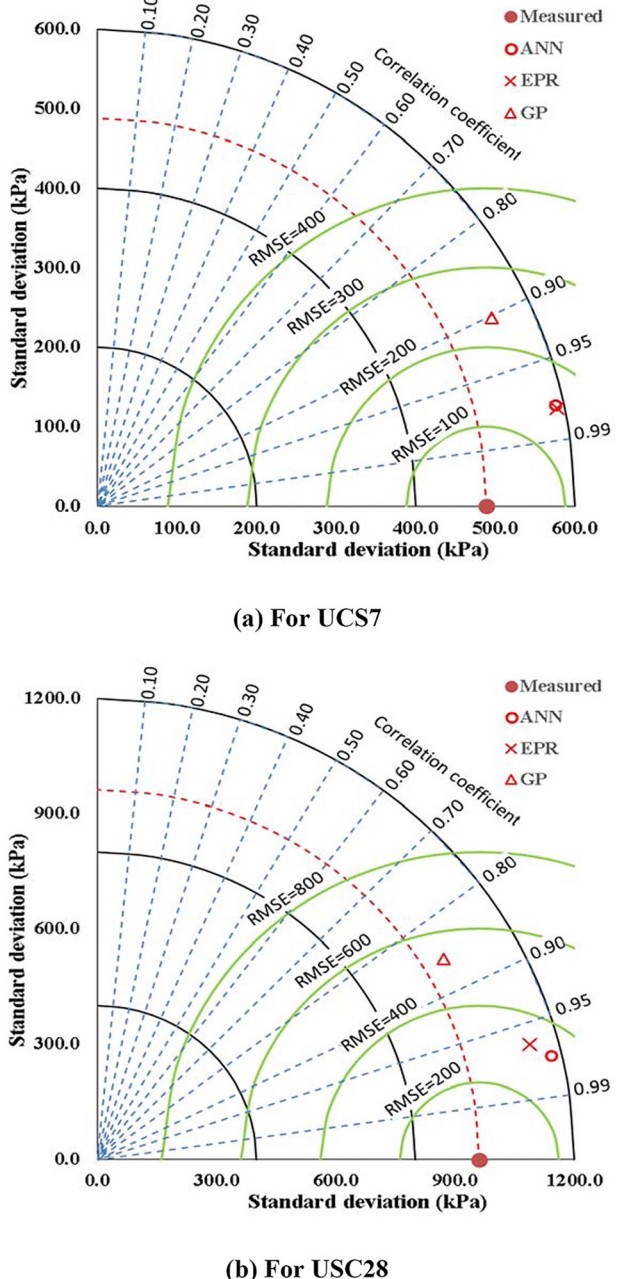

**Fig 10.** Comparing the accuracies of the developed models using Taylor charts, a) For UCS7, b) For UCS28.

can be used to navigate the design space. The Predicted $R^2$ for the UCS 28 of 0.9942 is in reasonable agreement with the Adjusted $R^2$ of 0.8780; i.e., the difference is less than 0.2. Adeq precision measures the signal to noise ratio. A ratio greater than 4 is desirable. Your ratio of 14.722 indicates an adequate signal. This model can be used to navigate the design space. These are presented in Table 7. The UCS7 equation (Eq 8) in terms of actual factors can be used to make predictions about the UCS response for given levels of each factor with high accuracy and adequate precision. Here, the levels should be specified in the original units for each factor. This UCS7 equation should not be used to determine the relative impact of each

**Table 7. Fit Statistics of the RSM UCS model.**

| UCS 7 | | | | UCS 28 | | | |
|---|---|---|---|---|---|---|---|
| Std. Dev. | 317.33 | $R^2$ | 0.8427 | Std. Dev. | 626.70 | $R^2$ | 0.8405 |
| Mean | 861.82 | Adjusted $R^2$ | 0.8805 | Mean | 1440.07 | Adjusted $R^2$ | 0.8780 |
| C.V. % | 36.82 | Predicted $R^2$ | 0.9879 | C.V. % | 43.52 | Predicted $R^2$ | 0.9942 |
| | | Adeq Precision | 14.1694 | | | Adeq Precision | 14.7220 |

factor because the coefficients are scaled to accommodate the units of each factor and the intercept is not at the center of the design space. The UCS28 equation (Eq 9) in terms of actual factors can be used to make predictions about the lime reconstituted soil strength response for given levels of each factor with high performance accuracy. Here, the levels should be specified in the original units for each factor. This equation should not be used to determine the relative impact of each factor because the coefficients are scaled to accommodate the units of each factor and the intercept is not at the center of the design space.

$$
\begin{aligned}
UCS7 = {}& 3.00273E + 06 - 43948.95160G - 60393.75334S - 61832.70889M \\
& - 59877.68708C + 322.80660L + 441.16380G*S + 453.56275G*M \\
& + 435.09326G*C - 4.91665G*L + 621.52991S*M + 602.89345S*C - 1.95523S*L \\
& + 616.86171M*C - 1.66066M*L - 0.774502C*L + 162.15468G^2 + 303.45791S^2 \\
& + 318.08864M^2 + 298.26931C^2 - 9.66276L^2
\end{aligned} \tag{8}
$$

$$
\begin{aligned}
UCS28 = {}& 5.41923E + 06 - 66782.14211G - 1.09543E + 05S - 1.11595E + 05M \\
& - 1.07794E + 05C - 1082.29002L + 670.25613G*S + 694.67228G*M \\
& + 655.88448G*C + 11.98451G*L + 1127.09317S*M + 1090.27190S*C \\
& + 14.79881S*L + 1110.45194M*C + 15.30650M*L + 16.88140C*L \\
& + 171.27991G^2 + 553.64498S^2 + 574.04265M^2 + 535.51876C^2 - 26.49127L^2
\end{aligned} \tag{9}
$$

The optimized model representation are shown in Figs 11–21 and these succinctly illustrate the normal plot of residuals for normal percentage probability and externally studentized residuals for the unconfined compressive strength RSM model, the illustrative representation

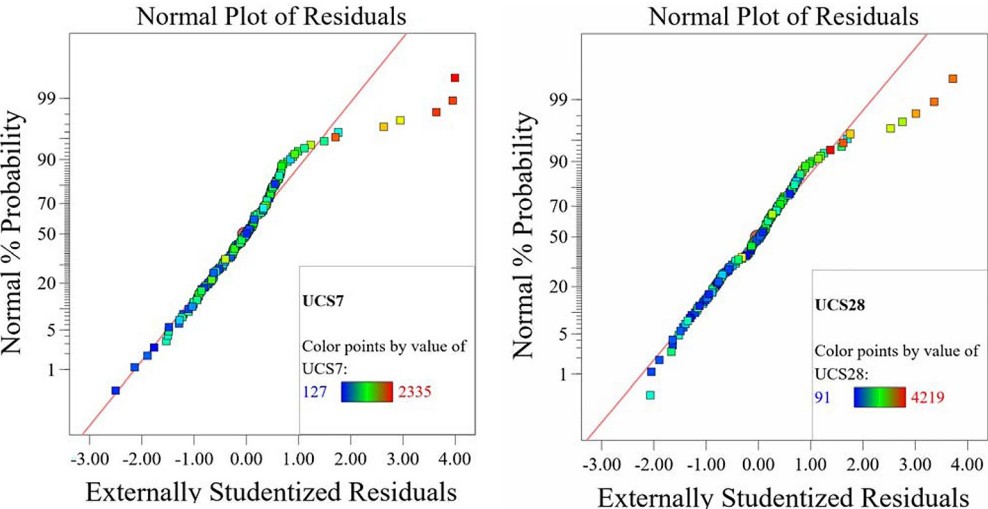

**Fig 11. Normal plot of residuals for normal percentage probability and externally studentized residuals for the unconfined compressive strength RSM model.**

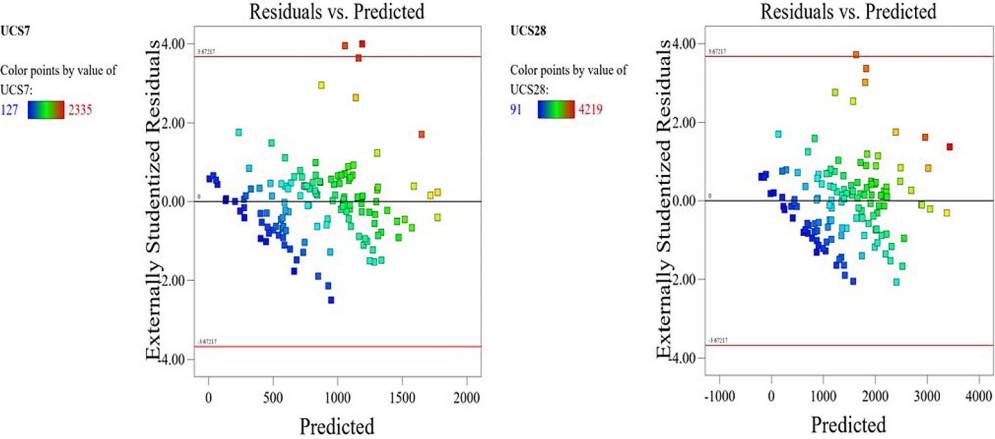

**Fig 12. Illustrative representation of the externally studentized residuals and the predicted values of the UCS model.**

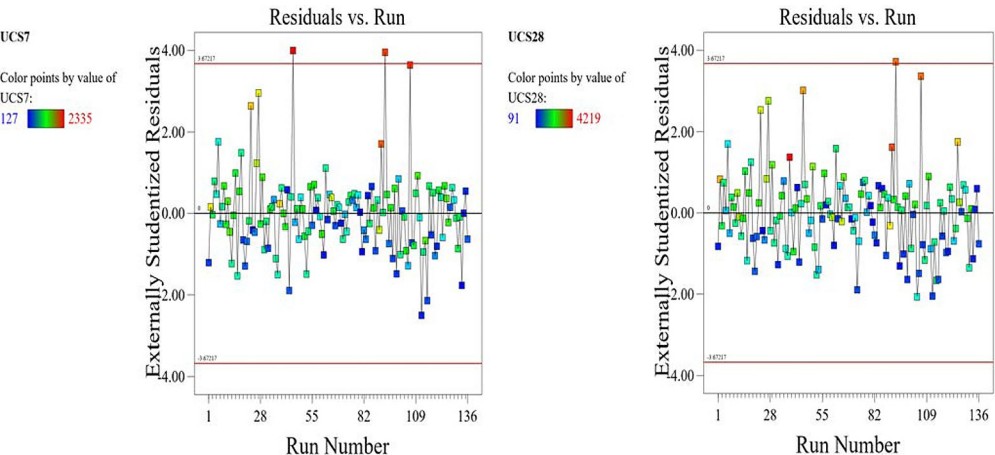

**Fig 13. Illustrative representation of the externally studentized residuals and the entry runs of the UCS model.**

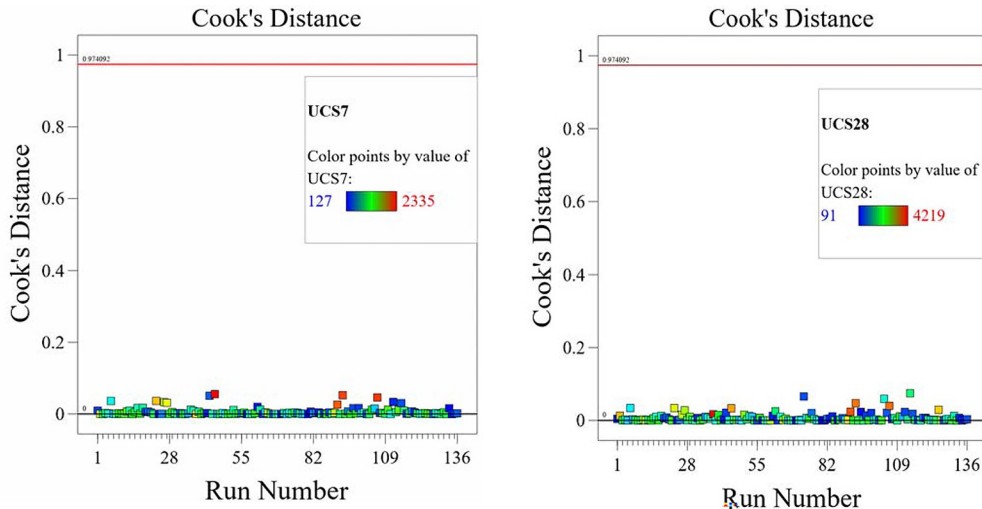

**Fig 14. Illustrative representation of the Cook's distance of the UCS model.**

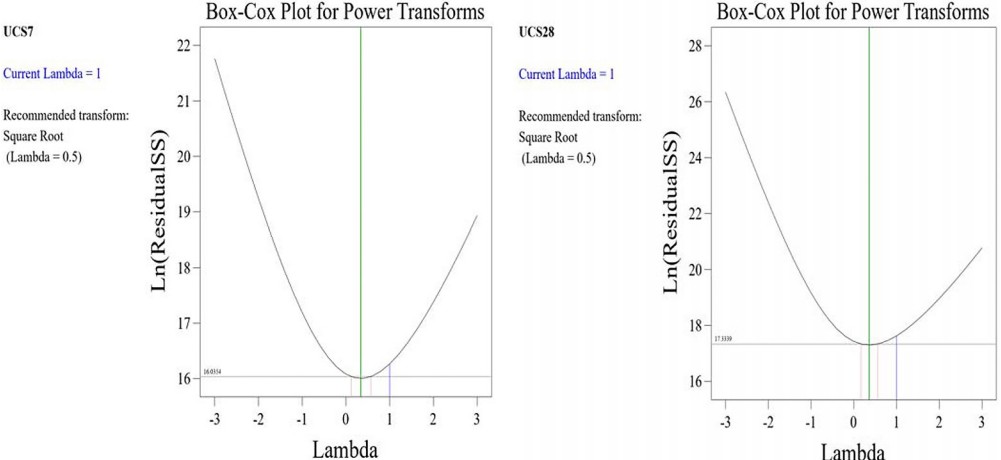

**Fig 15. Illustrative representation of the Box-Cox Plot for Power Transform of the UCS model.**

of the externally studentized residuals and the predicted values of the UCS model, the externally studentized residuals and the entry runs of the UCS model, the Cook's distance of the UCS model, the Box-Cox Plot for Power Transform, the predicted versus actual values of the UCS model, the residuals versus gravel parameter values of the UCS model, the leverage versus run values of the UCS model, the DFFITS versus run values of the UCS model, the DFBETAS versus run values of the UCS model, and the perturbation, FDS, and 3D surface behavior of the UCS model with selected parameters, respectively. These further show the optimization strengths of the RSM to produce graphical presentations of the behavior of the modeled unconfined compressive strength of the lime reconstituted soil for the purpose of the sustainable design and construction subgrade and landfill line for a sustainable environmental safety. These further show the point at which selected parameters of this study is optimized against the value of the UCS at both 7 and 28 days curing regime. These performance agrees with

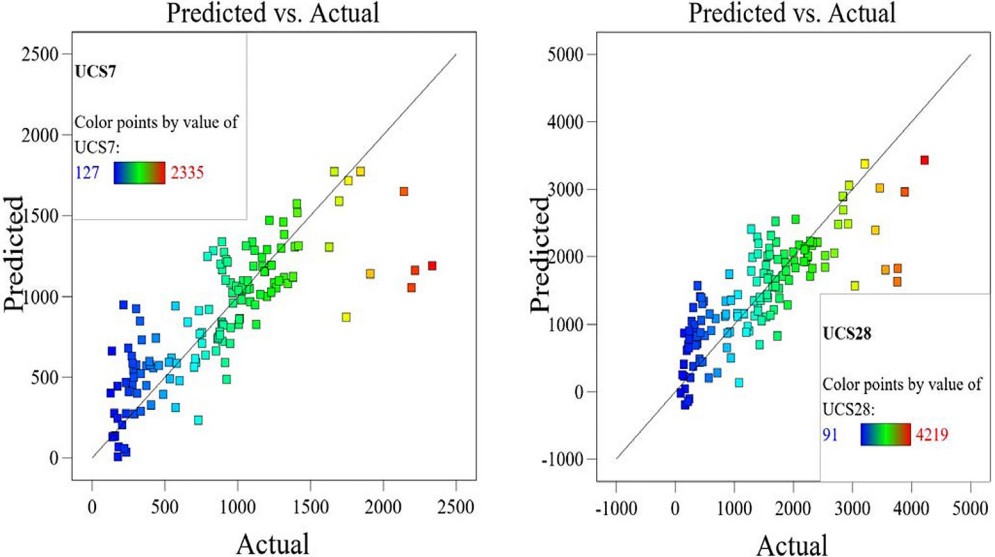

**Fig 16. Illustrative representation of the predicted versus actual values of the UCS model.**

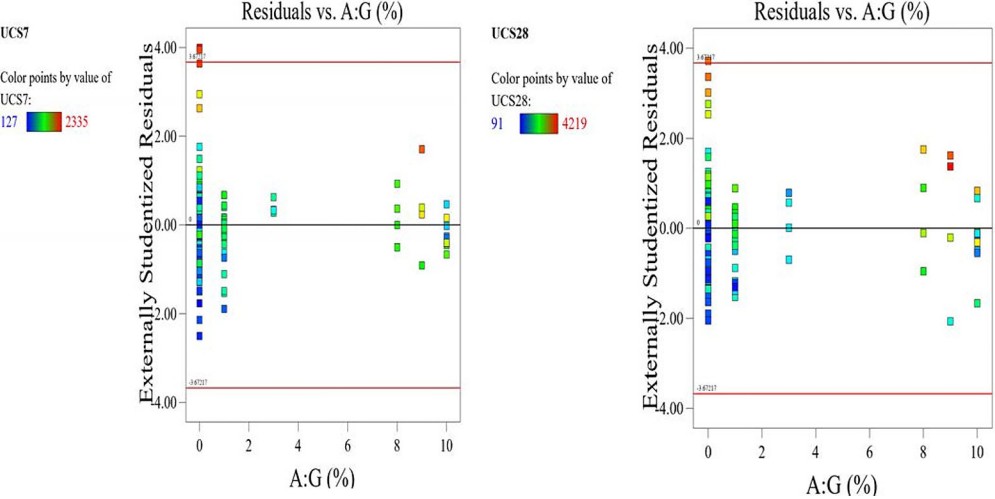

**Fig 17. Illustrative representation of the residuals versus gravel parameter values of the UCS model.**

previous works where RSM had been applied as the decisive model [52–55]. Comparatively, the perfromance level of the RSM is to be considwred seriosly in the deisgn and cinstruction of civil engineering structures like the road pavement subgrade and landfill liner due to its ability to propose a closed-form equation at over 95% accuracy and over 14% adequate precision, that could be applied manually and automatically during the design and constructted structures performance monitoring [49,50]. In Fig 21, a 3D surface and FDS graphical behavior of the UCS was presented showing the 3D behavioral effect of selected parameters. It further shows the behavioral consistency between the UCS at 28 days of the lime reconstituted soil and the G versus S, M, and C proportions in the overall mixes. It can be adduced that the UCS improved with higher values of S, M, and C against higher values of the G, which produced a reduction trend in the behavior of the UCS at 28 days.

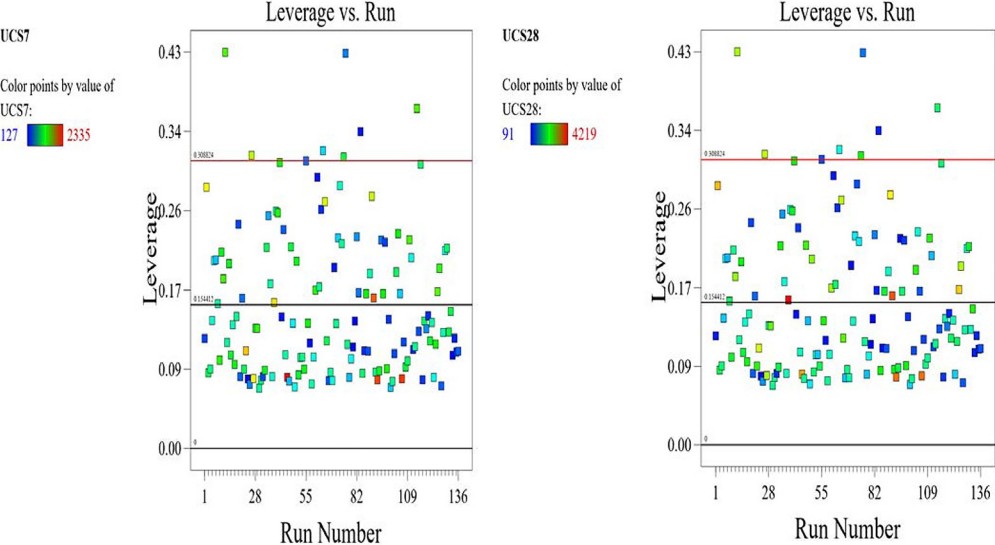

**Fig 18. Illustrative representation of the leverage versus run values of the UCS model.**

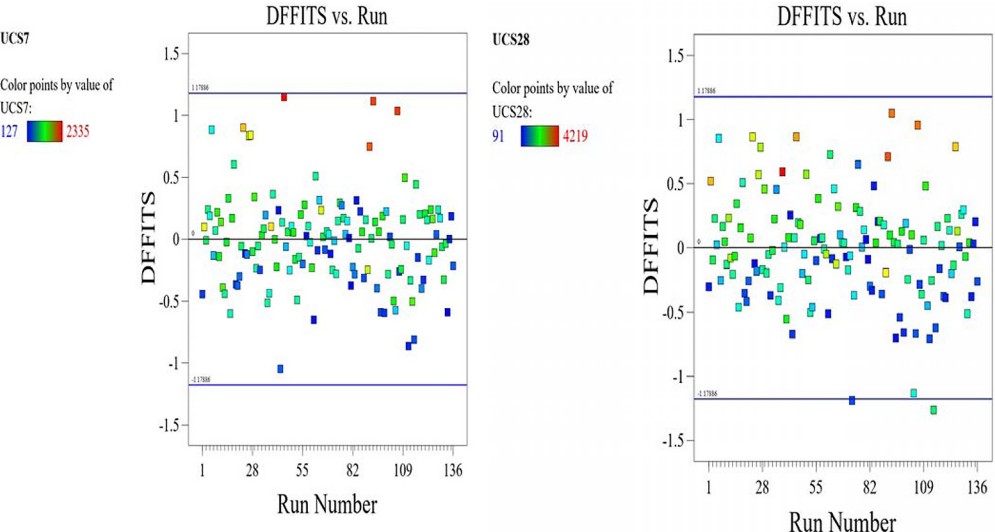

**Fig 19. Illustrative representation of the DFFITS versus run values of the UCS model.**

## 5. Conclusions

This study presents three models utilizing Artificial Intelligence (AI) techniques, namely Genetic Programming (GP), Artificial Neural Network (ANN), and Evolutionary Polynomial Regression (EPR), and one utilizing a symbolic algorithm known as the Response Surface Methodology (RSM) to predict the unconfined compressive strength after 7 and 28 days (UCS7 and UCS28 in kPa) for lime-stabilized soil. The models incorporate input variables such as Gravel, Sand, Silt, Clay, and Lime contents (G, S, M, C, L). The following key conclusions can be drawn from comparing the accuracies of the developed models:

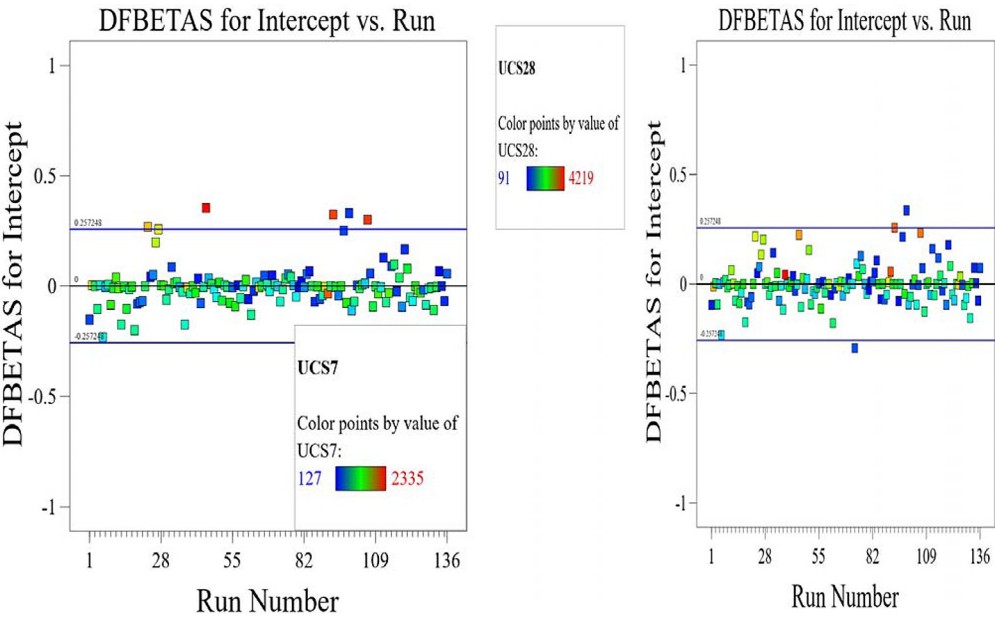

**Fig 20. Illustrative representation of the DFBETAS versus run values of the UCS model.**

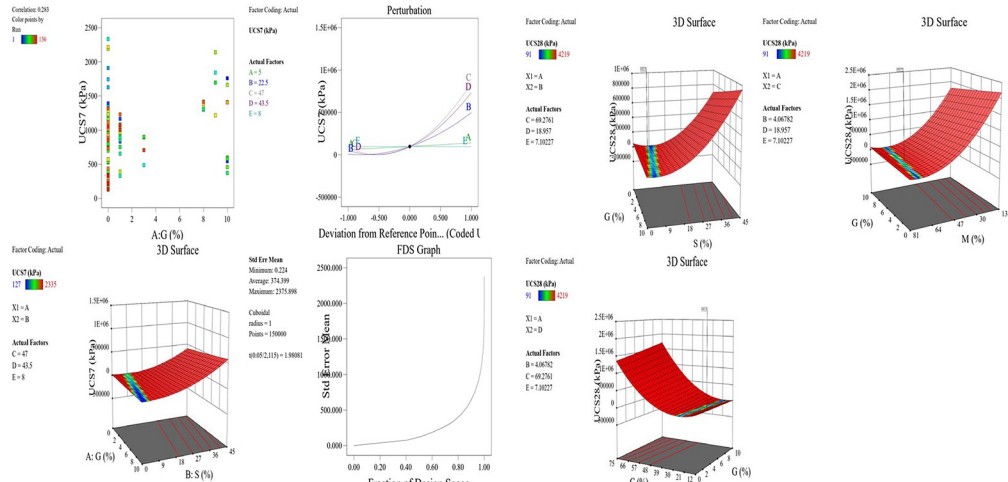

**Fig 21. Illustrative representation of the perturbation, FDS, and 3D surface behavior of the UCS model with selected parameters.**

- Both ANN and EPR demonstrated similar levels of accuracy, reaching approximately 88% for the compressive strength after 7 days (UCS7), while the GP model exhibited a lower accuracy level of 76%.

- The prediction accuracy of the compressive strength after 28 days (UCS28) was lower compared to UCS7 due to the increased complexity of the required formula. The accuracy for ANN and EPR models was approximately 85% and 82%, respectively. Conversely, the GP model exhibited the lowest performance with an accuracy of 66.0%.

- The results indicate that all factors have nearly equal importance for both UCS7 and UCS28, except for the lime content (L), which has a lesser influence.

- Similar to other regression techniques, the generated formulas are valid within the considered range of parameter values. Beyond this range, it is essential to verify the prediction accuracy.

- However, the RSM model compared well with the results of the AI-powered models in performance accuracy and further proposed a closed-form equation for manual and automatic application in the design of optimized utilization of lime in soil treated for the purpose of subgrade and landfill liner construction. This is very important in practice as quicker manual applications are needed to verify optimized material usage for the best performance.

- This research emphasizes the immense potential of ML techniques in predicting the unconfined compressive strength of lime reconstituted graded soil mixtures. It further emphasises the impactful influence of the finer soils in the sand and clay categories on the strength of the studied lime reconstituted soil. The findings contribute to a better understanding of the behavior of lime-stabilized soil, offering valuable insights for engineering applications in the field of soil mechanics. However, the results and validity of the models are within the selected database and the soil treated with lime. So, future research work is expected to extend towards studying other sustainable cementitious materials database applied in soil for the design and construction of subgrade and landfill liners.

## Author Contributions

**Conceptualization:** Xinghuang Guo, Kennedy Onyelowe.

**Data curation:** Cesar Garcia, Alexis Ivan Andrade Valle, Kennedy Onyelowe, Ahmed M. Ebid.

**Formal analysis:** Xinghuang Guo, Cesar Garcia, Alexis Ivan Andrade Valle, Kennedy Onyelowe, Andrea Natali Zarate Villacres, Ahmed M. Ebid.

**Funding acquisition:** Xinghuang Guo, Alexis Ivan Andrade Valle.

**Investigation:** Cesar Garcia, Andrea Natali Zarate Villacres.

**Methodology:** Alexis Ivan Andrade Valle, Kennedy Onyelowe, Shadi Hanandeh.

**Project administration:** Xinghuang Guo, Shadi Hanandeh.

**Resources:** Xinghuang Guo, Ahmed M. Ebid, Shadi Hanandeh.

**Software:** Kennedy Onyelowe, Ahmed M. Ebid.

**Supervision:** Xinghuang Guo, Kennedy Onyelowe.

**Validation:** Xinghuang Guo, Cesar Garcia, Kennedy Onyelowe, Andrea Natali Zarate Villacres.

**Visualization:** Alexis Ivan Andrade Valle.

**Writing – original draft:** Xinghuang Guo, Cesar Garcia, Alexis Ivan Andrade Valle, Kennedy Onyelowe, Andrea Natali Zarate Villacres, Ahmed M. Ebid, Shadi Hanandeh.

**Writing – review & editing:** Xinghuang Guo, Andrea Natali Zarate Villacres.

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
