## [Decision Letter · Decision Letter 0]

21 Feb 2024

PONE-D-24-03879Modeling the influence of lime on the unconfined compressive strength of reconstituted G-S-M-C graded soil using advanced machine learning approaches for subgrade and liner applicationsPLOS ONE

Dear Dr. Onyelowe,

Thank you for submitting your manuscript to PLOS ONE. After careful consideration, we feel that it has merit but does not fully meet PLOS ONE’s publication criteria as it currently stands. Therefore, we invite you to submit a revised version of the manuscript that addresses the points raised during the review process.

We look forward to receiving your revised manuscript.

Kind regards,

Dr. S. M. Anas, Ph.D.(Structural Engg.), M.Tech(Earthquake Engg.)

Academic Editor

PLOS ONE

Journal Requirements:

2. In your Methods section, please provide additional information regarding the permits you obtained for the work. Please ensure you have included the full name of the authority that approved the field site access and, if no permits were required, a brief statement explaining why

4. "In this instance it seems there may be acceptable restrictions in place that prevent the public sharing of your minimal data. However, in line with our goal of ensuring long-term data availability to all interested researchers, PLOS’ Data Policy states that authors cannot be the sole named individuals responsible for ensuring data access (http://journals.plos.org/plosone/s/data-availability#loc-acceptable-data-sharing-methods).

Additional Editor Comments:

Dear Authors,

I hope this email finds you well. I am writing to inform you about the decision regarding your manuscript entitled "Modeling the influence of lime on the unconfined compressive strength of reconstituted G-S-M-C graded soil using advanced machine learning approaches for subgrade and liner applications" (PONE-D-24-03879), which you submitted to PLOS ONE.

After careful consideration, I am pleased to inform you that your manuscript has undergone peer review by three experts in the field. However, all reviewers have provided feedback indicating that major revisions are required before the manuscript can be considered for publication. Based on their recommendations and our preliminary analysis, we have decided to request major revisions from you.

Please note that upon resubmission, your revised manuscript will undergo further evaluation to ensure that the necessary revisions have been adequately addressed. We encourage you to be thorough and transparent in your revisions to expedite the review process.

Important note from Academic Editor, Dr. S. M. Anas: -

I would like to bring to your attention that citing the papers suggested by the reviewers is not mandatory for your revised manuscript. It is entirely up to you whether or not you choose to include the suggested papers in your revised version. The reviewers have provided these suggestions to enhance the quality and credibility of your research, but ultimately, the decision is yours. You have the freedom to decline including any of the suggested papers in your revised manuscript if you feel they are not relevant or do not add value to your study.

If you have any questions or need clarification on any aspect of the revision process, please do not hesitate to contact me. We appreciate your commitment to enhancing the quality of your manuscript and thank you for choosing PLOS ONE as the outlet for your research.

We look forward to receiving your revised manuscript.

Best regards,

Dr. S. M. Anas

Academic Editor

PLOS ONE

Reviewers' comments:

Reviewer's Responses to Questions

**Comments to the Author**

1. Is the manuscript technically sound, and do the data support the conclusions?

Reviewer #1: Yes

Reviewer #2: Yes

Reviewer #3: Yes

2. Has the statistical analysis been performed appropriately and rigorously? 

Reviewer #1: Yes

Reviewer #2: Yes

Reviewer #3: Yes

3. Have the authors made all data underlying the findings in their manuscript fully available?

Reviewer #1: Yes

Reviewer #2: Yes

Reviewer #3: Yes

4. Is the manuscript presented in an intelligible fashion and written in standard English?

Reviewer #1: Yes

Reviewer #2: Yes

Reviewer #3: Yes

5. Review Comments to the Author

Reviewer #1: Dear Authors

Manuscript entitled” Modeling the influence of lime on the unconfined compressive strength of reconstituted G-S-M-C graded soil using advanced machine learning approaches for subgrade and liner applications “

This manuscript with Manuscript Number PONE-D-24-03879 as been written very great, and it has good novelty for advanced machine learning approaches for subgrade and liner applications of Grout, therefor I am some suggestions to improve your manuscript.

1- the title of the paper is not informative and not clear, so I suggest to the authors to revise the title.

2- The research contributions of the paper should be articulated more clearly. The abstract is not representative of the content and contributions of the paper. The abstract does not seem to properly convey the rigor of research. And also I recommend the author(s) to be more direct to cover the research gap, manuscript's goal, materials and methods, main results, main contributions, main suggestions for future research.

3- The necessity and innovation of the article should be presented to the introduction.

4- In addition, there author should highlight the advantages of which on advanced machine learning approaches in introduction section. Literature review section does not present relevant and updated works that form an adequate basis for the study's research model. The author could add a table in Introduction section and list download the more recent studies used advanced machine learning approaches for soil improvement.

5- Following, you will find some new related references which should be added to literature review:

• Salehi, M., Bayat, M., Saadat, M., & Nasri, M. (2023). Prediction of unconfined compressive strength and California bearing capacity of cement-or lime-pozzolan-stabilised soil admixed with crushed stone waste. Geomechanics and Geoengineering, 18(4), 272-283.

• Karimiazar, J., Teshnizi, E. S., O'Kelly, B. C., Sadeghi, S., Karimizad, N., Yazdi, A., & Arjmandzadeh, R. (2023). Effect of nano-silica on engineering properties of lime-treated marl soil. Transportation Geotechnics, 43, 101123. https://doi.org/10.1016/j.trgeo.2023.101123

• Baldovino, J. A., Moreira, E. B., Izzo, R. L. D. S., & Rose, J. L. (2018). Empirical relationships with unconfined compressive strength and split tensile strength for the long term of a lime-treated silty soil. Journal of Materials in Civil Engineering, 30(8), 06018008. https://doi.org/10.1061/(ASCE)MT.1943-5533.000237

• Hashemi, H., Yazdi, A., & Teshnizi, E. S. (2023). Improvement of collapsing problematic soils on the sabzevar-mashhad railway route (northeast of Iran) using traditional additives. Nexo Revista Científica, 36(03), 383-403. https://doi.org/10.5377/nexo.v36i03.16461

• Goutham, D. R., & Krishnaiah, A. J. (2024). Prediction of unconfined compressive strength of expansive soil amended with bagasse ash and lime using artificial neural network. Journal of Soft Computing in Civil Engineering, 8(1), 33-54.

• Sharifi Teshnizi, E., Mirzababaei, M., Karimiazar, J., Arjmandzadeh, R., & Mahmoudpardabad, K. (2024). Gypsum and rice husk ash for sustainable stabilization of forest road subgrade. Quarterly Journal of Engineering Geology and Hydrogeology, 57(1), qjegh2023-008. https://doi.org/10.1144/qjegh2023-00

• Sharma, L. K., & Singh, T. N. (2018). Regression-based models for the prediction of unconfined compressive strength of artificially structured soil. Engineering with computers, 34, 175-186. https://doi.org/10.1007/s00366-017-0528-8

6- For readers to quickly catch your contribution, it would be better to highlight major difficulties and challenges, and your original achievements to overcome them, in a clearer way in abstract and introduction.

7- It is suggested to compare the results of the present research with some similar studies which is done before.

8- Much more explanations and interpretations must be added for the Results, that's not enough.

9- Some key parameters are not mentioned. The rationale on the choice of the particular set of parameters should be explained with more details. Have the authors experimented with other sets of values? What are the sensitivities of these parameters on the results?

10- Why do you choose only graded soil (G,S,M, and C) and Lime percent in this research? You know type of clay minerals and reaction to lime additive for improving of soil are really important, how can you effect it in your models?

11- You should add the flow chart in this manuscript for methodology.

12- Why did you manually separate the training and Validation datasets? Couldn't they be automatically separated and selected by artificial intelligence models?

13- Have you controlled these models in other additives?

14- In the tables of training and Validation datasets, are some UCS in 7 days more than 28 days? What was the reason for this?

15- The conclusion section would be improved by starting with the research goal. The author could present the practical implication of this study. However, further suggestions for future research could be addressed and inserting a paragraph by stating the limitations of this study.

Thanks

Reviewer #2: The work titled "Modeling the Influence of Lime on the Unconfined Compressive Strength of Reconstituted Gravel-Sand-Silt-Clay (G-S-M-C) Graded Soil Using Advanced Machine Learning Approaches for Subgrade and Liner Applications" presents valuable insights. However, to enhance the paper, the following comments should be addressed:

1. Title: The title of the paper is commendably descriptive, but its length may pose readability challenges. Additionally, the use of uncommon abbreviations such as G-S-M-C might deter readers unfamiliar with geotechnical engineering terminology. A concise and accessible title without abbreviations would enhance readability and broaden the paper's appeal.

2. Abstract: The abstract serves as a crucial gateway to the paper's content. To improve its effectiveness, consider reorganizing it to follow a logical structure that highlights the main findings of the research. Structuring the abstract to include sections for Introduction, problem statement, methodology, results, and conclusion would provide readers with a clear roadmap of the paper's contents. Furthermore, replacing the abbreviation G-S-M-C with a more commonly understood term would enhance accessibility to a broader audience within geotechnical engineering.

3. Introduction: While the introduction effectively sets the stage for the study, providing additional background information could strengthen its impact. Including more context about the research problem and its significance would help readers better understand the motivation behind the study. Moreover, concluding the introduction with a paragraph emphasizing the importance, novelty, and originality of the research would further engage readers and highlight the paper's contributions to the field.

4. Data Collection and Statistical Analysis: Presenting the data collection process in a tabular format would enhance clarity and facilitate easier interpretation for readers. Additionally, avoiding the use of abbreviations in subsection titles would contribute to a more professional presentation of the research.

5. Figures 7 and 8: Adding a line Y=X to these figures would improve clarity and aid in understanding the data presented.

6. RSM Model Analysis: Considering the complexity of equations 8 and 9, relocating them to an appendix rather than the main section of the paper would streamline the presentation and improve readability.

7. Results: In addition to reporting positive findings, it's essential to discuss any unexpected or negative results encountered during the study. Comparing the results with those of previous studies would provide valuable context and highlight the novelty of the findings.

8. Conclusions: Streamlining the conclusions to focus on key findings and their scientific implications would enhance clarity and impact. Furthermore, adding closing remarks summarizing all conclusive points would provide a cohesive ending to the paper and reinforce its scientific contributions.

Reviewer #3: The paper “Modeling the influence of lime on the unconfined compressive strength of reconstituted G-S-M-C graded soil using advanced machine learning approaches for subgrade and liner applications” has been thoroughly studied and reviewed. This paper has presented four machine learning (ML) techniques, such as Genetic Programming (GP), Artificial Neural Networks (ANN), Evolutionary Polynomial Regression (EPR), and the Response Surface Methodology in modeling the unconfined compressive strength (UCS) of soil-lime mixtures for subgrade and landfill liner design and construction. This is one of the best model presentations I have reviewed recently, which proposed a very important concept in solving environmental problems related to highway foundations and landfill liners. The authors have done excellently well in both English language presentation, study background, methodology, results presentation and analyses and this promises to spur and open up research projects in this area. The conclusions are supported by the results of the research work. Meanwhile, there exist very minor revisions needed to update the already established standard of research work;

1. What could be responsible for the low performance of the GP compared to the other models?

2. What is the reason behind the excellent performance of the RSM as a symbolic regression technique?

3. The practical application of the symbolic closed-form equations 3, 4, 6-9 is required to be mentioned in the results discussion section.

With these comments addressed in the manuscript, the research paper stands out as an excellent work in the field of engineering and technology in solving geo-environmental engineering problems.

6. PLOS authors have the option to publish the peer review history of their article (what does this mean?). If published, this will include your full peer review and any attached files.

Reviewer #1: **Yes: **Ebrahim Sharifi Teshnizi

Reviewer #2: **Yes: **Meysam Bayat

Reviewer #3: **Yes: **Mahmood Ahmad

---

## [Author Response · Author response to Decision Letter 0]

27 Feb 2024

PLOS ONE

Reply to comments

Paper ID: PONE-D-24-03879

Paper Title: Modeling the influence of lime on the unconfined compressive strength of reconstituted G-S-M-C graded soil using advanced machine learning approaches for subgrade and liner applications

Reviewer #1: Dear Authors

Manuscript entitled” Modeling the influence of lime on the unconfined compressive strength of reconstituted G-S-M-C graded soil using advanced machine learning approaches for subgrade and liner applications “

This manuscript with Manuscript Number PONE-D-24-03879 as been written very great, and it has good novelty for advanced machine learning approaches for subgrade and liner applications of Grout, therefor I am some suggestions to improve your manuscript.

1- the title of the paper is not informative and not clear, so I suggest to the authors to revise the title.

R1: This has been revised as required.

2- The research contributions of the paper should be articulated more clearly. The abstract is not representative of the content and contributions of the paper. The abstract does not seem to properly convey the rigor of research. And also I recommend the author(s) to be more direct to cover the research gap, manuscript's goal, materials and methods, main results, main contributions, main suggestions for future research.

R2: The abstract section has been adjusted to meet the concerns raised here, thank you, professor for these insights.

3- The necessity and innovation of the article should be presented to the introduction.

R3: This important part of the work has been updated as required.

4- In addition, there author should highlight the advantages of which on advanced machine learning approaches in introduction section. Literature review section does not present relevant and updated works that form an adequate basis for the study's research model. The author could add a table in Introduction section and list download the more recent studies used advanced machine learning approaches for soil improvement.

R4: The presently listed reviewed papers contain the information needed in their contents and could occupy unnecessary space if listed as in a tabular form in the present paper with its present volume. The readers can refer to those listed literatures for more details if needed. I hope this helps. 

5- Following, you will find some new related references which should be added to literature review:

• Salehi, M., Bayat, M., Saadat, M., & Nasri, M. (2023). Prediction of unconfined compressive strength and California bearing capacity of cement-or lime-pozzolan-stabilised soil admixed with crushed stone waste. Geomechanics and Geoengineering, 18(4), 272-283.

• Karimiazar, J., Teshnizi, E. S., O'Kelly, B. C., Sadeghi, S., Karimizad, N., Yazdi, A., & Arjmandzadeh, R. (2023). Effect of nano-silica on engineering properties of lime-treated marl soil. Transportation Geotechnics, 43, 101123. https://doi.org/10.1016/j.trgeo.2023.101123

• Baldovino, J. A., Moreira, E. B., Izzo, R. L. D. S., & Rose, J. L. (2018). Empirical relationships with unconfined compressive strength and split tensile strength for the long term of a lime-treated silty soil. Journal of Materials in Civil Engineering, 30(8), 06018008. https://doi.org/10.1061/(ASCE)MT.1943-5533.000237

• Hashemi, H., Yazdi, A., & Teshnizi, E. S. (2023). Improvement of collapsing problematic soils on the sabzevar-mashhad railway route (northeast of Iran) using traditional additives. Nexo Revista Científica, 36(03), 383-403. https://doi.org/10.5377/nexo.v36i03.16461

• Goutham, D. R., & Krishnaiah, A. J. (2024). Prediction of unconfined compressive strength of expansive soil amended with bagasse ash and lime using artificial neural network. Journal of Soft Computing in Civil Engineering, 8(1), 33-54.

• Sharifi Teshnizi, E., Mirzababaei, M., Karimiazar, J., Arjmandzadeh, R., & Mahmoudpardabad, K. (2024). Gypsum and rice husk ash for sustainable stabilization of forest road subgrade. Quarterly Journal of Engineering Geology and Hydrogeology, 57(1), qjegh2023-008. https://doi.org/10.1144/qjegh2023-00

• Sharma, L. K., & Singh, T. N. (2018). Regression-based models for the prediction of unconfined compressive strength of artificially structured soil. Engineering with computers, 34, 175-186. https://doi.org/10.1007/s00366-017-0528-8

R5: The most relevant from the suggested literature, which we found helpful have been cited and referenced in the manuscript texts. So, we thank the reviewer for the suggestions. 

6- For readers to quickly catch your contribution, it would be better to highlight major difficulties and challenges, and your original achievements to overcome them, in a clearer way in abstract and introduction.

R6: These have been revised accordingly as required. Thank you. 

7- It is suggested to compare the results of the present research with some similar studies which is done before.

R7: This has been updated further in the overall discussion of the results as required. Thank you for this insightful comment.

8- Much more explanations and interpretations must be added for the Results, that's not enough.

R8: More explanations have been added as required in the results discussion sections. 

9- Some key parameters are not mentioned. The rationale on the choice of the particular set of parameters should be explained with more details. Have the authors experimented with other sets of values? What are the sensitivities of these parameters on the results?

R9: the studied and mentioned parameters are already contained the United States soil stabilization database and you don’t expect the authors to fabricate what was contained in the original database or wrongly manipulate values. The methodology used in this research paper included data gathering, sorting and application in a machine learning application to model the strength of a lime treated soil. I hope you understand this, better. 

10- Why do you choose only graded soil (G,S,M, and C) and Lime percent in this research? You know type of clay minerals and reaction to lime additive for improving of soil are really important, how can you effect it in your models?

R10: This model project for your information in case you didn’t read it well is based on an already deposited database taken from the United State soil stabilization database as shown and cited in the data collection section. So, I expect that you do not need to make such comments as though it was an experiment conducted in my lab. And I don’t expect you to have me fabricate what is or are not contained in an original database thereby falsifying records to satisfy your concern. I hope you understand. Thank you. 

11- You should add the flow chart in this manuscript for methodology.

R11: The flowchart has been added as needed. 

12- Why did you manually separate the training and Validation datasets? Couldn't they be automatically separated and selected by artificial intelligence models?

R12: The database was automatically and randomly partitioned, sir, using the selected percentage for training and validation based on already established standards from research investigations.

13- Have you controlled these models in other additives?

R13: What exactly is your concern here? These ML methods have been used in other databases of concrete and soil with different components mixes and not exactly as used here. If that’s your concern. The additive used in this research paper is obviously lime as you can you see mentioned everywhere in the manuscript texts. Considering other additives in this work would mean going to the lab to deign mixes and proportions of such additives to be used to treat graded soil. I hope you understand what you are asking, prof. Thank you. 

14- In the tables of training and Validation datasets, are some UCS in 7 days more than 28 days? What was the reason for this?

R14: The training set for 7 days is equal to the training set for 28 days cured mix contained in Table 1, please check well. And same is the case for the validation sets for 7 and 28 days cured mixes contained in Table 2. Thank you. 

15- The conclusion section would be improved by starting with the research goal. The author could present the practical implication of this study. However, further suggestions for future research could be addressed and inserting a paragraph by stating the limitations of this study.

R15: This has been updated as needed. Thank you for the insight, sir. 

Reviewer #2: The work titled "Modeling the Influence of Lime on the Unconfined Compressive Strength of Reconstituted Gravel-Sand-Silt-Clay (G-S-M-C) Graded Soil Using Advanced Machine Learning Approaches for Subgrade and Liner Applications" presents valuable insights. However, to enhance the paper, the following comments should be addressed:

1. Title: The title of the paper is commendably descriptive, but its length may pose readability challenges. Additionally, the use of uncommon abbreviations such as G-S-M-C might deter readers unfamiliar with geotechnical engineering terminology. A concise and accessible title without abbreviations would enhance readability and broaden the paper's appeal.

R1: This has been revised as it was equally mentioned by reviewer #1. Thank you prof. 

2. Abstract: The abstract serves as a crucial gateway to the paper's content. To improve its effectiveness, consider reorganizing it to follow a logical structure that highlights the main findings of the research. Structuring the abstract to include sections for Introduction, problem statement, methodology, results, and conclusion would provide readers with a clear roadmap of the paper's contents. Furthermore, replacing the abbreviation G-S-M-C with a more commonly understood term would enhance accessibility to a broader audience within geotechnical engineering.

R2: The authors also made sure to define what that acronym means at its first use in the abstract and hope that the readers take a clue from the there. Thank you so much.

3. Introduction: While the introduction effectively sets the stage for the study, providing additional background information could strengthen its impact. Including more context about the research problem and its significance would help readers better understand the motivation behind the study. Moreover, concluding the introduction with a paragraph emphasizing the importance, novelty, and originality of the research would further engage readers and highlight the paper's contributions to the field.

R3: This has been updated as mentioned. Thank you. 

4. Data Collection and Statistical Analysis: Presenting the data collection process in a tabular format would enhance clarity and facilitate easier interpretation for readers. Additionally, avoiding the use of abbreviations in subsection titles would contribute to a more professional presentation of the research.

R4: These concerns have been worked on and eliminated as required. 

5. Figures 7 and 8: Adding a line Y=X to these figures would improve clarity and aid in understanding the data presented.

R5: The modeling about finding the best line of fit from the training and validation data entries presented in the scatter plots of Fig. 7 and 8, which have changed to Fig. 8 and 9. And, the best lines of fit for y = Ax have been presented in the graphs, which produced the final determination coefficient and it would have been unnecessary and clumsy to add a line that’s not relevant to the results of the models. I hope it makes clearer sense to you, sir. 

6. RSM Model Analysis: Considering the complexity of equations 8 and 9, relocating them to an appendix rather than the main section of the paper would streamline the presentation and improve readability.

R6: Equations 8 and 9 are closed-form equations from a symbolic regression model, the RSM and its presentation as a main content in the manuscript text should not be considered an issue to readability, rather a consistent attempt to at a glance make reference to a mentioned equation in the text. It would have been clumsy and redundant if the formulation of the equations were manual the whole step of which is presented in a manuscript. So, Prof, you see we do not have a problem allowing the equations like the others within the text, ok. Thank you prof. 

7. Results: In addition to reporting positive findings, it's essential to discuss any unexpected or negative results encountered during the study. Comparing the results with those of previous studies would provide valuable context and highlight the novelty of the findings.

R7: You are very correct about reporting negative outcomes but in this case, the present research has no such to report and as such didn’t report any. Thank you for your insights. 

8. Conclusions: Streamlining the conclusions to focus on key findings and their scientific implications would enhance clarity and impact. Furthermore, adding closing remarks summarizing all conclusive points would provide a cohesive ending to the paper and reinforce its scientific contributions.

R8: This has been updated as required, Professor. Thank you. 

Reviewer #3: The paper “Modeling the influence of lime on the unconfined compressive strength of reconstituted G-S-M-C graded soil using advanced machine learning approaches for subgrade and liner applications” has been thoroughly studied and reviewed. This paper has presented four machine learning (ML) techniques, such as Genetic Programming (GP), Artificial Neural Networks (ANN), Evolutionary Polynomial Regression (EPR), and the Response Surface Methodology in modeling the unconfined compressive strength (UCS) of soil-lime mixtures for subgrade and landfill liner design and construction. This is one of the best model presentations I have reviewed recently, which proposed a very important concept in solving environmental problems related to highway foundations and landfill liners. The authors have done excellently well in both English language presentation, study background, methodology, results presentation and analyses and this promises to spur and open up research projects in this area. The conclusions are supported by the results of the research work. Meanwhile, there exist very minor revisions needed to update the already established standard of research work;

1. What could be responsible for the low performance of the GP compared to the other models?

R1: This is because they depend on three partially or fully stochastic (i.e., randomly determined) operations, which are selection, crossover and mutation and therefore may not find an optimal solution in a reasonable amount of time. 

2. What is the reason behind the excellent performance of the RSM as a symbolic regression technique?

R2: Because it can be used to determine the interaction effects of the independent input parameters. The data-driven model equation can be utilized to illustrate the different combinations of independent input factors that affect the outcome of a process/product.

3. The practical application of the symbolic closed-form equations 3, 4, 6-9 is required to be mentioned in the results discussion section.

R3: This has been updated in the discussion section as required. 

With these comments addressed in the manuscript, the research paper stands out as an excellent work in the field of engineering and technology in solving geo-environmental engineering problems.

The authors wish to thank the reviewers for their time and insightful comments and references suggestions, which have added value and quality to the present research paper. We are most grateful and this to the cause of research. Thank you.

---

## [Decision Letter · Decision Letter 1]

11 Mar 2024

Modeling the influence of lime on the unconfined compressive strength of reconstituted graded soil using advanced machine learning approaches for subgrade and liner applications

PONE-D-24-03879R1

Dear Dr. Onyelowe,

We’re pleased to inform you that your manuscript has been judged scientifically suitable for publication and will be formally accepted for publication once it meets all outstanding technical requirements.

Kind regards,

Dr. S. M. Anas, Ph.D.(Structural Engg.), M.Tech(Earthquake Engg.)

Academic Editor

PLOS ONE

Additional Editor Comments (optional):

Dear Authors,

I hope this email finds you well. I am writing to inform you about the decision on your revised manuscript entitled "Modeling the influence of lime on the unconfined compressive strength of reconstituted graded soil using advanced machine learning approaches for subgrade and liner applications" (PONE-D-24-03879R1), which has undergone further review.

I am pleased to inform you that the revised manuscript was sent to the previous reviewers for reevaluation. I am delighted to convey that all three reviewers have recommended the paper for publication and expressed satisfaction with the authors' responses to their comments.

Based on the recommendations of the reviewers and a preliminary assessment of the revised manuscript, I have decided to take an Accept decision, subject to the approval of the editorial board.

Once again, congratulations on the successful revision of your manuscript. I will keep you updated on any further progress regarding the editorial board's decision.

Thank you for your contribution to PLOS ONE.

Best regards,

Dr. S. M. Anas

Academic Editor

PLOS ONE

Reviewers' comments:

Reviewer's Responses to Questions

**Comments to the Author**

1. If the authors have adequately addressed your comments raised in a previous round of review and you feel that this manuscript is now acceptable for publication, you may indicate that here to bypass the “Comments to the Author” section, enter your conflict of interest statement in the “Confidential to Editor” section, and submit your "Accept" recommendation.

Reviewer #1: All comments have been addressed

Reviewer #2: All comments have been addressed

Reviewer #3: All comments have been addressed

2. Is the manuscript technically sound, and do the data support the conclusions?

Reviewer #1: Yes

Reviewer #2: Yes

Reviewer #3: Yes

3. Has the statistical analysis been performed appropriately and rigorously? 

Reviewer #1: Yes

Reviewer #2: N/A

Reviewer #3: Yes

4. Have the authors made all data underlying the findings in their manuscript fully available?

Reviewer #1: Yes

Reviewer #2: Yes

Reviewer #3: Yes

5. Is the manuscript presented in an intelligible fashion and written in standard English?

Reviewer #1: Yes

Reviewer #2: Yes

Reviewer #3: Yes

6. Review Comments to the Author

Reviewer #1: The authors correlate the Modeling the influence of lime on the unconfined compressive strength of reconstituted graded soil using advanced machine learning approaches for subgrade and liner applications from the literature. The study could be of importance to the future readers who would want to use these correlations in their study.

This manuscript was improved for all comments.

Thanks

Reviewer #2: The new version of the paper is wholly modified compared to the original version, and the article is acceptable for publication.

Reviewer #3: Most of the reviewer concerns are addressed so recommended to the Editor Highness for acceptance of this manuscript.

7. PLOS authors have the option to publish the peer review history of their article (what does this mean?). If published, this will include your full peer review and any attached files.

Reviewer #1: No

Reviewer #2: No

Reviewer #3: **Yes: **Mahmood Ahmad

---

## [Editor Report · Acceptance letter]

18 Mar 2024

PONE-D-24-03879R1 

PLOS ONE

Dear Dr. Onyelowe, 

I'm pleased to inform you that your manuscript has been deemed suitable for publication in PLOS ONE. Congratulations! Your manuscript is now being handed over to our production team.

Kind regards, 

on behalf of

Dr. S. M. Anas 

Academic Editor

PLOS ONE